# Emergence of Exploration in Policy gradient reinforcement learning via Retrying

**Soichiro Nishimori** [1]  **Paavo Parmas** [1]  **Sotetsu Koyamada** [2 3 4]
**Tadashi Kozuno** [5]  **Toshinori Kitamura** [6]  **Shin Ishii** [3 4]  **Yutaka Matsuo** [1]

## Abstract

In reinforcement learning (RL), agents benefit from exploration *only* because they repeatedly encounter similar states: trying different actions can then improve performance or reduce uncertainty; without such retries, a greedy policy is optimal. We formalize this intuition with **Re-Max**, an objective that evaluates a policy by the expected maximum return over $M$ samples ($M \in \mathbb{N}$), while accounting for return uncertainty. Optimizing this objective induces stochastic exploration as an emergent property, without explicit bonus terms. For efficient policy optimization, we derive a new policy-gradient formulation for ReMax and introduce **ReMax PPO** (**RePPO**), a PPO variant that optimizes ReMax while generalizing the discrete retry count $M$ to a continuous parameter $m > 0$, enabling fine-grained control of exploration. Empirically, RePPO promotes exploration—without any explicit exploration bonuses—on the MinAtar and Craftax benchmarks. The official code is available at https://github.com/nissymori/remax-rl.

## 1. Introduction

Exploration is a central problem in reinforcement learning (RL) and has been extensively studied (Sutton & Barto, 2018; Haarnoja et al., 2018; Ziebart et al., 2008; Mnih et al., 2016). A prevailing approach adds bonuses, such as entropy (Haarnoja et al., 2018; Ziebart et al., 2008) or count-based bonuses (Bellemare et al., 2016; Ostrovski et al., 2017), to the environment's reward to encourage exploration explicitly. Other lines of work instantiate

---
[1]The University of Tokyo, Tokyo, Japan [2]Kobe University, Kobe, Japan [3]Kyoto University, Kyoto, Japan [4]ATR, Kyoto, Japan [5]Isara Laboratories, Inc., Tokyo, Japan [6]University of Alberta, Edmonton, Canada. Correspondence to: Soichiro Nishimori <nishimori@ms.k.u-tokyo.ac.jp>, Paavo Parmas <paavo.parmas@weblab.t.u-tokyo.ac.jp>.

*Proceedings of the 43rd International Conference on Machine Learning*, Seoul, South Korea. PMLR 306, 2026. Copyright 2026 by the author(s).

posterior sampling (Thompson, 1933) using ensembles or Bayesian networks (Osband et al., 2016; 2018; 2019). We study a distinct mechanism that drives exploration via greedy reward maximization.

The objective of RL is not to maximize the reward on the current trial, but rather to learn to maximize the rewards after several trials. Exploration matters because it enables achieving higher rewards on subsequent trials. Without the opportunity for retrying, exploration is unnecessary: the rational choice is the action currently believed to yield the highest reward. Environmental uncertainty also motivates exploration, encouraging attempts at alternative actions for information gathering. If there is no uncertainty, we do not need to explore: the problem is reduced to pure optimization.

Building on this principle—*decision-makers retry under uncertainty*—we propose the **ReMax** objective, which formalizes exploration as reward maximization under uncertainty. We first present ReMax in a bandit setting and compare it to the standard RL objective.

**The ReMax objective (bandit).** Let $K \geq 2$ be the number of actions, let $\mu = (\mu_1, \ldots, \mu_K)$ denote per-action values, and let $\Pi$ be a distribution over $\mu$ (e.g., the agent's current belief/posterior over unknown values). For a policy $\pi \in \Delta^{K-1}$ ($\Delta^{K-1}$ is the probability simplex over $K$ actions), $M \in \mathbb{N}$ and $[M] \coloneqq \{1, \ldots, M\}$,

$$J_{\mathrm{RL}}(\pi) \coloneqq \mathbb{E}_{A \sim \pi} [\mu_A],$$
$$J_{\mathrm{ReMax}}^M(\pi) \coloneqq \mathbb{E}_{\boldsymbol{\mu} \sim \boldsymbol{\Pi}} \left[ \mathbb{E}_{A_{[M]} \sim \boldsymbol{\pi}} \left[ \max_{m \in [M]} \mu_{A_m} \mid \mu \right] \right]. \quad (1)$$

Blue captures **retrying**: we score a policy by the best of $M$ draws. For $M = 1$ with fixed rewards, $J_{\mathrm{ReMax}}^M$ reduces to $J_{\mathrm{RL}}(\pi)$ and the optimum is deterministic (Sutton & Barto, 2018); for $M \geq 2$ it can be stochastic (Sec. 2). Red captures (epistemic) **uncertainty** (Ghosh et al., 2021) over $\mu$—uncertainty due to limited data rather than inherent randomness—which evolves during exploration; we address it via explicit modeling (Osband et al., 2016) or by sampling from nonstationary return estimates (Moalla et al., 2024).

Comparatively, one canonical approach to exploration is to augment the reward with curiosity-based bonuses, including pseudo-count methods (Bellemare et al., 2016; Lobel

et al., 2023) and prediction-error-based approaches (Pathak et al., 2017; Burda et al., 2019). While these methods have proven effective in ALE game domains, they typically require additional models to estimate the statistics needed to construct the bonuses, increasing algorithmic complexity and imposing extra computational overhead.

Unlike methods that add explicit bonuses, ReMax induces exploration *without* bonuses by optimizing a purely reward-based objective. Recent subsequent/concurrent works to ours, in LLM training for reasoning tasks have studied retry-style objectives such as pass@$K$ (Walder & Karkhanis, 2025; Tang et al., 2025b), which share a similar idea to ours. [1] The key distinction is that ReMax explicitly accounts for reward uncertainty, whereas LLM reasoning tasks typically assume fixed, verifiable rewards. For a broader discussion and connections to related work, see App. A.

Throughout this paper, we address the following question:

> Can we promote exploration without adding explicit bonuses by optimizing ReMax in RL?

To answer this question, we organize the paper as follows.

**Step 1: Empirical study in Bandits.** We empirically illustrate the core idea of how ReMax (defined in Eq. (1)) induces effective exploration in bandits in Sec. 2. As the retry parameter $M$ increases, the optimal policy becomes more exploratory, and ReMax adapts exploration to the scale of reward uncertainty; in a posterior bandit setting, it exhibits *empirically* sublinear regret (as observed in our experiments).

**Step 2: ReMax in RL.** We define the ReMax objective for RL in Sec. 3. Unlike bandits, state transitions hinder retrying multiple actions from the same state to observe returns, so we emulate retries via queries to a $Q$-function and discuss possible instantiations of ReMax in RL.

**Step 3: Policy Gradient for ReMax.** To optimize ReMax, we develop a practical policy-gradient (PG) method in Sec. 4. We derive a new PG formulation that is estimable from trajectory returns and generalize the integer draw count $M$ to a positive real parameter $m > 0$, enabling finer control of the exploration–exploitation trade-off. Building on this formulation, we introduce **ReMax PPO** (RePPO), a PPO-based deep actor–critic algorithm (Schulman et al., 2017).

**Step 4: Experiments.** Finally, we evaluate RePPO on MinAtar (Young & Tian, 2019) and Craftax (Matthews et al., 2024) in Sec. 5. RePPO optimizes ReMax without exploration bonuses, achieves better performance, and

maintains higher policy entropy than PPO with an entropy bonus; peak performance occurs around $m = 1.2$–$1.4$. On Craftax, a larger-scale open-ended RL environment, RePPO achieves competitive performance compared to a tuned entropy-regularized PPO, despite not using an exploration bonus. Overall, ReMax emerges as a promising objective for exploration in reinforcement learning.

## 2. ReMax in Bandits: An Empirical Study

This section builds intuition for how ReMax (Eq. (1)) promotes exploration via retrying and uncertainty. We first design simple reward distributions to illustrate ReMax's optima (Sec. 2.1), then move to a posterior-updating bandit where $\Pi$ is learned from data (i.e., a standard Bayesian bandit learning setting as in Thompson sampling; Sec. 2.2).

### 2.1. Warm-up: Properties of ReMax.

**ReMax optima yield stochastic policies.** This example shows how retrying can induce stochastic policies. Consider a two-armed bandit with arms indexed by $a \in \{0, 1\}$ (so $\mu = (\mu_0, \mu_1)$): $\mu = (0, 1)$ w.p. 0.75 and $\mu = (1, 0)$ w.p. 0.25. For the RL objective, the optimal policy is deterministic and always chooses arm 1. For ReMax (Eq. (1)), the optimal policy is stochastic: mixing between arms hedges which one is rewarding (repeating the same arm cannot improve the max-over-$M$ value). Since $\mu = (0, 1)$ is more likely, a limited retry budget $M$ still assigns substantial mass to arm 1 to avoid missing it.

As $M$ grows, the policy can explore arm 0 more often and the optimum becomes increasingly exploratory. Fig. 1 (Left) plots $J_{\text{ReMax}}^M(p)$ vs. $p := \pi(a{=}1)$ for $M = 1, \ldots, 5$ (analytic values; see App. C.1). Dots mark the maximizer $p^*$: $p^* = 1$ for $M = 1$ (value 0.75), and for $M \geq 2$ it shifts toward exploration as the value approaches 1. Therefore, the retry mechanism induces stochastic behavior in the presence of uncertainty.

**ReMax adapts exploration to reward uncertainty.** The previous example showed that ReMax induces stochastic behavior by the retry mechanism; here we show that it adapts to the magnitude of reward uncertainty. We consider a two-armed Bernoulli bandit with $\mu_i = \alpha_i X_i$, where $X_i \sim$ Bernoulli($p_i$) and $\mathbb{E}[\mu_i] = \alpha_i p_i$. We fix $p_0 = 1$ and $\alpha_0 = 2$, and vary $\alpha_1$ from 1 to 10, adjusting $p_1$ so that $\alpha_1 p_1 = 1$ remains constant (fixed mean, varying variance). Fig. 1 (Center) shows the optimal probability of selecting arm 1, $\pi^*(a = 1)$, for $M = 2$, alongside that of the softmax policy, which is the analytical optimum of the RL objective with an entropy bonus (App. C.2). When $\alpha_1 \leq 2$, arm 1 is never chosen since its maximum cannot exceed that of arm 0. As $\alpha_1$ increases beyond 2, ReMax increasingly favors arm 1, reflecting its adaptiveness to rare but high-reward outcomes.

[1] See (Koyamada et al., 2022) for an early preprint of the current paper that proposed the ReMax objective and its simple optimization with REINFORCE already in 2022, preceding later max@$K$/pass@$K$ policy-optimization works in LLM training.

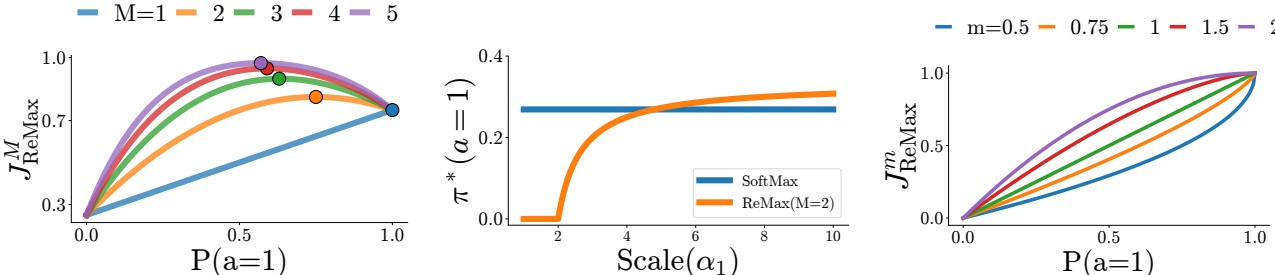

*Figure 1.* Bandit problems. (**Left**) ReMax objective for a binary bandit. The standard reinforcement learning objective ($M = 1$) has a deterministic optimal policy ($p^* = 1$); in contrast, increasing the retry count $M \geq 2$ shifts the optimal policy toward stochastic exploration to hedge against reward uncertainty. (**Center**) A plot of how the optimal policy changes as the reward variance changes. While Softmax remains fixed regardless of variance, ReMax automatically increases exploration as reward uncertainty (scale $\alpha_1$) grows, targeting rare high-reward outcomes. (**Right**) ReMax objective for a fixed deterministic reward vector. The continuous parameter $m$ reshapes the objective's curvature. Values of $m > 1$ flatten the gradient to slow convergence and sustain exploration, while $m < 1$ accelerates updates.

In contrast, Softmax remains flat across $\alpha_1 \in [1, 10]$ as it depends solely on the mean, which is fixed.

**ReMax with deterministic rewards.** The previous examples focused on ReMax with stochastic rewards. We now turn to the deterministic setting and show that, *even with fixed rewards*, ReMax reshapes the objective geometry and thus modulates convergence via the retry parameter.

Consider a binary bandit with rewards fixed to $\mu = (0, 1)$ and let $p := \pi(a=1) \in [0, 1]$. In this case, the ReMax objective admits a closed form: for $m > 0$, $J_{\text{ReMax}}^m(p) = 1 - (1 - p)^m$, where $m$ can be an integer or a positive real. Fig. 1 (Right) plots $J_{\text{ReMax}}^m(p)$ for $m = 0.5, 0.75, 1, 1.5, 2$, foreshadowing our continuous-$m$ formulation in Sec. 3 and Sec. 4. The maximizer remains $p^\star = 1$ for all $m$, indicating that after sufficient exploration removes epistemic uncertainty, the policy will converge to the optimal policy. However, the local geometry near high $p$ depends strongly on $m$: larger $m$ flattens the objective and reduces gradient magnitudes, whereas smaller $m$ sharpens curvature and amplifies gradients. Thus, tuning $m$ controls convergence even in deterministic settings: $m > 1$ slows updates (encouraging exploration), while $m < 1$ accelerates them, mitigating the slow convergence often observed with softmax policies (Hennes et al., 2020). Furthermore, non-integer $m$ naturally interpolates between integer retry counts (e.g., $m = 1.5$ fits between $m = 1$ and $m = 2$), enabling finer-grained control.

### 2.2. Bandit with Posterior: Empirical Sublinear Regret.

In the previous section, we intentionally designed the distribution over reward $\Pi$ to illustrate properties of ReMax. In practice, the distribution is *estimated* and updated from observed data as the agent explores the environment. To validate ReMax in this more realistic setting, we consider a $K$-armed bandit with posterior updates, optimize ReMax using samples from the evolving posterior (Thompson, 1933),

and evaluate cumulative regret.

**Problem setup.** At each run, draw means $(\mu_1, \ldots, \mu_K) \sim \Pi^*$ and fix them, with $\mu^* = \max_i \mu_i$. At round $t \in [T]$, we choose $A_t$, observe $R_t$ (mean $\mu_{A_t}$), and update the posterior $\Pi_{t+1}$ (prior $\Pi_0 = \Pi^*$). ReMax optimizes $\widehat{J}_{\text{ReMax}}^M(\pi_\theta)$ using samples from $\Pi_t$ to produce $\pi_t$ (see App. C.3). We study (i) **Beta–Bernoulli** ($\Pi_0 = \text{Beta}(1, 1)$, $R_t \sim \text{Bernoulli}(\mu_{A_t})$) and (ii) **Gaussian–Gaussian** ($\Pi_0 = \mathcal{N}(0, 1)$, $R_t \sim \mathcal{N}(\mu_{A_t}, 1)$). We compare with Thompson sampling (Thompson, 1933; Agrawal & Goyal, 2017; Honda & Takemura, 2014), UCB (Auer et al., 2002) (both sublinear-regret), and a Softmax baseline, where we took the softmax of the posterior means and selected the arm following the distribution. We use $K = 10$, $T = 1000$, and $M \in \{2, 3\}$, and report mean $\pm$ standard error cumulative regret over 256 runs (instantaneous regret $\mu^* - \mu_{A_t}$); full details are in App. C.3.

**Results.** ReMax exhibits empirically sublinear cumulative regret, comparable to the classic UCB and Thompson sampling baselines, while Softmax incurs higher cumulative regret, especially for the Gaussian-Gaussian bandit (Fig. 2). We do not claim superiority; rather, these results demonstrate that ReMax yields effective exploration in practice. We leave theoretical regret bounds to future work.

## 3. ReMax in RL

We extend ReMax from bandits to episodic Markov Decision Processes (MDPs) (Puterman, 2014) $\mathcal{M} = (\mathcal{S}, \mathcal{A}, r, P, T)$ with discrete actions $\mathcal{A} = \{1, \ldots, K\}$. For $\tau \sim (\pi, P)$, let $\mathcal{R}(\tau) := \sum_{t=0}^{T-1} r(s_t, a_t)$ and maximize $\mathbb{E}[\mathcal{R}(\tau)]$ (subscript by $\mathcal{M}$, i.e., $\mathcal{R}_\mathcal{M}, \mathcal{Q}_\mathcal{M}$, when needed).

In extending ReMax from bandits to RL, two issues arise: the unit of **uncertainty** and the feasibility of **retries**. Uncertainty in RL concerns the entire environment, including rewards and transitions, so we place a distribution over

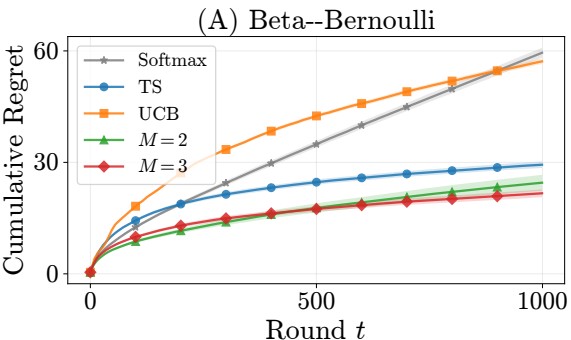
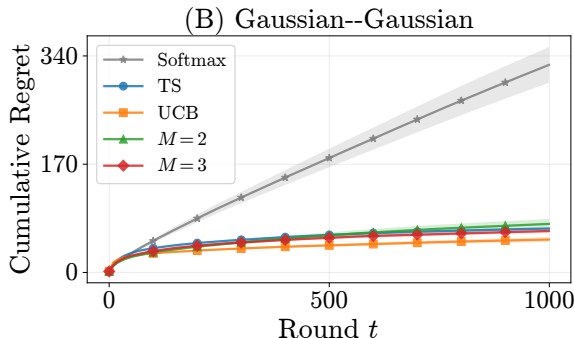

*Figure 2.* Average cumulative regret and standard error over 256 runs. ReMax with $M = 2$ and $M = 3$, optimized by gradient ascent.

MDPs, $\mathcal{M} \sim P(\mathcal{M})$, capturing (epistemic) uncertainty over unexplored regions of the environment (Ghosh et al., 2021). Retrying in RL would naïvely require multiple roll-outs from the same state until termination, which is infeasible without a resettable simulator (Ecoffet et al., 2021) and often prohibitively expensive even with one. To address this, we introduce function approximation via a Q-function $Q_{\mathcal{M}}^{\pi}(s, a) := \mathbb{E}_{\tau \sim (\pi, P)}[\mathcal{R}_{\mathcal{M}}(\tau) \mid s, a]$, i.e., the expected episodic return from starting in state $s$, taking action $a$, and then following $\pi$; this replaces Monte Carlo returns from $(s, a)$ with queries to $Q_{\mathcal{M}}^{\pi}$. To make the retry structure explicit, we fix a state $s \in \mathcal{S}$ at which retries are considered. In practice, $s$ ranges over states encountered during training; our actor–critic implementation optimizes an average of this objective over the on-policy state distribution. The ReMax objective is defined as

$$
\begin{aligned}
&J_{\text{ReMax}}^{M}(\pi, s) \\
&:= \mathbb{E}_{\mathcal{M} \sim P(\mathcal{M})}\left[\mathbb{E}_{A_{[M]} \sim \pi}\left[\max_{m \in [M]} Q_{\mathcal{M}}^{\pi}(s, A_m)\right]\right].
\end{aligned} \tag{2}
$$

As in the bandit setting, setting $M = 1$ with a fixed environment $\mathcal{M}$ recovers the standard RL objective, and the optimal policy is deterministic (Sutton & Barto, 2018). In Eq. (2), all within-environment randomness is already absorbed into $Q_{\mathcal{M}}^{\pi}$, so the dependence on $\mathcal{M}$ is only through $Q_{\mathcal{M}}^{\pi}$. Let $\mathcal{Q}$ denote the induced distribution over Q-functions (from $\mathcal{M} \sim P(\mathcal{M})$ or an uncertainty model); replacing the outer expectation over $\mathcal{M}$ by an expectation over $Q \sim \mathcal{Q}$ yields a practical form for optimization:

**Definition 3.1** (ReMax objective with Q distribution).

$$
\begin{aligned}
&J_{\text{ReMax}}^{M}(\pi, s, \mathcal{Q}) \\
&:= \mathbb{E}_{Q \sim \mathcal{Q}}\left[\mathbb{E}_{A_{[M]} \sim \pi}\left[\max_{m \in [M]} Q(s, A_m)\right]\right].
\end{aligned} \tag{3}
$$

For a fixed state, the Q-values $Q(s, \cdot)$ form a $K$-dimensional vector, reducing the problem to the bandit case. Therefore,

we can expect that ReMax enjoys the exploration advantages observed in Sec. 2. In practice, optimizing the ReMax objective involves several algorithmic considerations.

**Modeling Q-function uncertainty.** The objective in Eq. (3) can be estimated given samples of Q-values from the distribution $\mathcal{Q}$, via two approaches: (1) **Explicit modeling:** explicitly model $\mathcal{Q}$ with ensembles or Bayesian methods (Osband et al., 2016; 2018; 2019; An et al., 2021); model-based methods can likewise capture environment-level uncertainty (Hafner et al., 2020). (2) **Implicit modeling:** in standard deep RL, critics $Q_{\phi}(s, a)$ are inherently nonstationary due to distribution shift and algorithmic randomness (Moalla et al., 2024; Tang & Berseth, 2024), so we treat $Q_{\phi}(s, \cdot)$ at each update as a sample from this *implicit* distribution. As in the bandit setting (Sec. 2), ReMax is expected to adapt to this variability for exploration and further, even with nearly deterministic Q-values, ReMax with $M > 1$ can still promote exploration by slowing down the convergence as demonstrated in the deterministic bandit in Sec. 2.1.

**Computing the expected maximum over $M$ trials.** Suppose a sample of Q-values for a state $s$ is given by $q = (Q(s, a_1), \dots, Q(s, a_K))$ with $Q \sim \mathcal{Q}$ via either approach in the previous paragraph. The question is how to compute the inner expectation for a fixed $q$, defined as $J_{\text{ReMax}}^{M}(\pi, s, q) := \mathbb{E}_{A_{[M]} \sim \pi}[\max_{m \in [M]} q_{A_m}]$. Sampling $M$ actions yields an unbiased but high-variance estimator, so we provide a closed form.

**Proposition 3.2** (Closed-form expression of the inner expectation). *Given a Q-value vector $q \in \mathbb{R}^K$, sort it as $q_{(1)} \geq \dots \geq q_{(K)}$, breaking ties arbitrarily, with aligned policy masses $\pi_{(j)}$. Define $C_0 := 0$ and $C_j := \sum_{u=1}^{j} \pi_{(u)}$, $j = 1, \dots, K$. Then,*

$$
J_{\text{ReMax}}^{M}(\pi, s, q) = q_{(1)} + \sum_{j=1}^{K-1} \left(q_{(j+1)} - q_{(j)}\right)\left(1 - C_j\right)^{M}. \tag{4}
$$

Refer to App. B.1 for the proof. The computational cost is $O(K \log K)$ to sort $q$ and does not depend on $M$. This

closed-form expression allows us to calculate the inner expectation exactly, and it is differentiable with respect to $\pi$. Note that this relies on a (relatively small) discrete action space, where Q-values for all actions are available and we can sort them; for continuous or vast action spaces (e.g., language models), sample-based estimation is required.

**Base algorithm.** ReMax can be integrated into a broad family of RL algorithms that rely on Q-functions as critics. We highlight two classes: (1) **Actor–critic:** Actor–critic methods (Schulman et al., 2017; Haarnoja et al., 2018) are a natural fit since they already optimize policies with respect to critic signals. Thus, ReMax can be instantiated by replacing the policy-optimization module of the actor–critic algorithm by ReMax. (2) **Q-learning:** ReMax can also be combined with Q-learning variants (Mnih et al., 2013; Gallici et al., 2025; Vieillard et al., 2020), though it requires training a policy model in addition to the Q-function.

**Our instantiation.** Because ReMax is a new policy-optimization objective, we adopt a simple and efficient instantiation: an on-policy actor–critic method with implicit Q-uncertainty and closed-form computation. Specifically, we use PPO (Schulman et al., 2017) as the base method due to its strong performance with discrete actions. We do not use Eq. (4) directly; instead, we derive a closed-form policy gradient that similarly leverages sorting in Sec. 4. Other instantiations, such as explicit Q-distribution modeling, optimizing ReMax through Eq. (4), or integrating ReMax with other RL algorithms (e.g., Q-learning (Mnih et al., 2013)), are left for future work.

## 4. Policy Gradient in ReMax

To optimize the ReMax objective in Eq. (3) within on-policy actor–critic methods, we develop a practical policy gradient (PG) approach. Since only a single-trajectory return from $(s, a)$ is observable, we design a PG estimator based on such returns. In Sec. 4.1, we show that a naïve PG derivation is not directly estimable from trajectory returns and derive an estimation-friendly reformulation. We then provide a closed form of our proposed PG and generalize the number of draws to a positive real $m$ to enable fine-grained control over exploration in Sec. 4.2. Finally, we present an actor–critic instantiation, **Re**Max **PPO** (RePPO), based on Q-critic PPO (Schulman et al., 2017).

### 4.1. Estimation-Friendly Policy Gradient for ReMax

We seek an unbiased PG for ReMax that is estimable from single-trajectory returns; we first recall why standard RL admits the REINFORCE estimator (Williams, 1992).

**Policy Gradient in Standard RL.** Let $\pi_\theta : \mathcal{S} \times \mathcal{A} \to [0, 1]$ be a parametrized policy, and define $J_{\mathrm{RL}}(\pi_\theta, s) :=$

$\mathbb{E}_{\tau \sim (\pi_\theta, P)} [\mathcal{R}(\tau) \mid s]$. The policy gradient theorem (Sutton & Barto, 2018) gives

$$\nabla_\theta J_{\mathrm{RL}}(\pi_\theta, s) = \mathbb{E}_{a \sim \pi_\theta} [\nabla_\theta \log \pi_\theta(a \mid s) \, Q^{\pi_\theta}(s, a)]. \quad (5)$$

Because the outer expectation is over a single action $\mathbb{E}_{a \sim \pi_\theta}[\cdot]$, replacing $Q^{\pi_\theta}(s, a)$ by the trajectory return $\mathcal{R}(s, a)$ yields an unbiased REINFORCE estimator: $\hat{g}_{\mathrm{RL}} := \nabla_\theta \log \pi_\theta(a \mid s) \, \mathcal{R}(s, a)$.

**Problem with the naïve PG for ReMax.** For ReMax with fixed Q-values $J_{\mathrm{ReMax}}^M(\pi_\theta, s, q)$, where $q \in \mathbb{R}^K$, applying the policy gradient theorem yields

$$\nabla_\theta J_{\mathrm{ReMax}}^M(\pi_\theta, s, q)$$
$$= \mathbb{E}_{A_{[M]} \sim \pi_\theta} \left[ \left( \max_{m \in [M]} q_{A_m} \right) \sum_{m=1}^M \nabla_\theta \log \pi_\theta(A_m \mid s) \right]. \quad (6)$$

Unlike Eq. (5), the expectation is over $M$ actions, so an unbiased estimator would require observing returns for all $M$ sampled actions, which is infeasible in episodic RL. Moreover, $A_1, \ldots, A_M$ are coupled through the $\max$ operator, so a single-action expectation does not follow directly. We resolve this by introducing a baseline that decouples the max and enables a single-action expectation.

**Policy gradient via expected improvement.** Following (Tang et al., 2025b), for each $m$ in Eq. (6), we can insert a baseline $b_m$ that may depend on $(s, A_{-m})$ but not on $A_m$:

$$\nabla_\theta J_{\mathrm{ReMax}}^M(\pi_\theta, s, q) = \mathbb{E}_{A_{[M]} \sim \pi_\theta} \Bigg[$$
$$\sum_{m=1}^M \nabla_\theta \log \pi_\theta(A_m \mid s) \left( \max_{j \in [M]} (q_{A_j} - b_m) \right) \Bigg], \quad (7)$$

which preserves unbiasedness of the policy gradient because $\mathbb{E}_{A_m} [\nabla_\theta \log \pi_\theta(A_m \mid s) b_m] = 0$. We choose $b_m$ as $W_{-m} := \max\{q_{A_1}, \ldots, q_{A_{m-1}}, q_{A_{m+1}}, \ldots, q_{A_M}\}$, having

$$\max_{j \in [M]} (q_{A_j} - W_{-m}) = (q_{A_m} - W_{-m})_+,$$

where $(x)_+ = \max(x, 0)$ for $x \in \mathbb{R}$. Intuitively, $W_{-m}$ is the best value among the other $M-1$ sampled actions, so $(q_{A_m} - W_{-m})_+$ measures how much $A_m$ improves on that best alternative (clipped at zero). This turns the $\max$ into an action-specific term plus an "others" term, enabling the following single-action form:

**Proposition 4.1.** *Let* $W_{M-1} := \max\{q_{A_1}, \ldots, q_{A_{M-1}}\}$. *Then, we have*

$$\nabla_\theta J_{\mathrm{ReMax}}^M(\theta, s, q) = \mathbb{E}_{a \sim \pi_\theta} \Bigg[$$
$$M \, \nabla_\theta \log \pi_\theta(a \mid s) \, \mathbb{E}_{A_{[M-1]} \sim \pi_\theta} [(q_a - W_{M-1})_+] \Bigg]. \quad (8)$$

See App. B.2 for the proof. The blue term is the expected improvement of the $M$-th draw when that draw is fixed to action $a$: it compares $q_a$ to $W_{M-1}$, the max over the other $M-1$ draws. Crucially, we take the expectation over the other $M-1$ actions $A_{[M-1]} \sim \pi_\theta$, which turns the coupled max into a scalar weight that depends only on $a$, $\pi_\theta$, and $q$. This is the key step that yields a single-action expectation and makes the PG estimable from single-trajectory returns. Since this quantity is central to our reformulated policy gradient, we refer to it as **Expected Improvement (EI)**, borrowing terminology from Bayesian optimization (Jones et al., 1998)[2]. For a reference $R \in \mathbb{R}$, policy $\pi$, and Q-values $q \in \mathbb{R}^K$, define $\mathrm{EI}_M(R, \pi, q) := \mathbb{E}_{A_{[M-1]} \sim \pi} [(R - W_{M-1})_+]$. Then, we have the EI-based PG.

**Definition 4.2** (EI-based PG). *For fixed Q-values $q \in \mathbb{R}^K$, the EI-based PG is*

$$
\begin{aligned}
&\nabla_\theta J_{\mathrm{ReMax}}^M(\theta, s, q) \\
&= M \, \mathbb{E}_{a \sim \pi_\theta} [\nabla_\theta \log \pi_\theta(a \mid s) \, \mathrm{EI}_M(q_a, \pi_\theta, q)].
\end{aligned}
\tag{9}
$$

Taking the expectation over q values ($q \in \mathbb{R}^K$) will recover the PG for ReMax in Eq. (3). As desired, a single-trajectory return from $(s, a)$ then yields the estimator $\hat{g}_{\mathrm{ReMax}} := M \nabla_\theta \log \pi_\theta(a \mid s) \, \mathrm{EI}_M(R, \pi_\theta, q)$ with $R = \mathcal{R}(s, a)$. While (Walder & Karkhanis, 2025; Tang et al., 2025b) use related comparator baselines mainly for variance reduction in sampling-based estimators for retry-style objectives, our reformulation expresses the gradient in terms of the policy probabilities $\pi_\theta$ and is therefore estimable from single-trajectory returns. In the following section, we show that EI also admits a closed-form computation.

## 4.2. Efficient and Generalized Computation of EI

We provide a closed-form computation of the EI by leveraging the sorting of Q-values as in Prop. 3.2.

**Proposition 4.3** (Closed-form computation of EI). *Let $q \in \mathbb{R}^K$ be Q-values at a state, $\pi \in \Delta^{K-1}$ a policy, $R \in \mathbb{R}$ a reference, and $M \in \mathbb{N}$. Define $v_i := (R - q_i)_+$ and sort $q$ as $q_{(1)} \geq \cdots \geq q_{(K)}$, breaking ties arbitrarily, with aligned masses $\pi_{(j)}$. Define $C_0 := 0$ and $C_j := \sum_{u=1}^j \pi_{(u)}$, $j = 1, \ldots, K$. Then, we have*

$$
\mathrm{EI}_M(R; \pi, q) = v_{(1)} + \sum_{j=1}^{K-1} \left(v_{(j+1)} - v_{(j)}\right) \left(1 - C_j\right)^{M-1}.
\tag{10}
$$

See App. B.3 for the proof. This also costs $\mathcal{O}(K \log K)$ time. Although ReMax is motivated by an integer number

---

**Algorithm 1** RePPO
1: **repeat**
2:     Collect trajectories under $\pi_\theta$ and compute returns $R_t^\lambda$.
3:     For each $(s_t, a_t)$: form $q \leftarrow Q_\phi(s_t, \cdot)$; compute $R_+(t) := \mathrm{EI}_m(R_t^\lambda; \pi_\theta, q)$, $Q_+(s_t, a) = \mathrm{EI}_m(Q_\phi(s_t, a); \pi_\theta, q)$, and advantage $A_+(t) = R_+(t) - b_+(s_t)$.
4:     Update actor by PPO objective (Eq. (12)) with $A_+(t)$; update critic $Q_\phi$ toward $R_t^\lambda$.
5: **until** convergence

---

of draws $M$, Eq. (10) naturally extends to real $m > 0$ by replacing $(1 - C_j)^{M-1}$ with $(1 - C_j)^{m-1}$. For $m < 1$, this finite-valued extension requires $C_j < 1$ for all $j < K$, which holds for full-support policies; in our implementation, we additionally clip $1 - C_j$ from below for numerical stability as in App. D. We therefore define the **generalized EI**, $\mathrm{EI}_m(R; \pi, q)$, by substituting $m$ for $M$, enabling finer control of the exploration–exploitation trade-off. The closed form of ReMax with fixed Q-values (Eq. 4) is also valid for any real $m > 0$.

### 4.3. RePPO: Practical Policy Gradient for ReMax

We obtain a policy gradient that can be estimated directly from trajectory returns (Def. 4.2) and computed in closed form (Eq. (10)). Incorporating this into PPO (Schulman et al., 2017) leads to our new algorithm, **ReMax PPO** (RePPO). We begin by revisiting the PPO surrogate.

**PPO surrogate.** PPO collects trajectories under $\pi_{\mathrm{old}}$, computes $\lambda$-returns $R_t^\lambda$, and forms advantages $A(t) = R_t^\lambda - V_\phi(s_t)$. It then optimizes a clipped, importance-weighted surrogate with $r_\theta(t) = \pi_\theta(a_t \mid s_t)/\pi_{\mathrm{old}}(a_t \mid s_t)$ and clip $\varepsilon > 0$, yielding:

$$
\begin{aligned}
&\mathcal{L}_{\mathrm{PPO}}(\theta) \\
&:= \mathbb{E} \left[ \min \left(r_\theta(t) A(t), \, \mathrm{clip}(r_\theta(t), 1-\varepsilon, 1+\varepsilon) A(t)\right) \right].
\end{aligned}
\tag{11}
$$

**RePPO surrogate.** RePPO modifies PPO in two ways: (1) use a Q-critic $Q_\phi$; (2) use an EI-based advantage. Given $R_t^\lambda$, define $R_+(t) := \mathrm{EI}_m(R_t^\lambda, \pi_\theta, Q_\phi(s_t, \cdot))$ and $Q_+(s_t, a) := \mathrm{EI}_m(Q_\phi(s_t, a), \pi_\theta, q)$ with $q = Q_\phi(s_t, \cdot)$. Set $b_+(s_t) := \mathbb{E}_{a \sim \pi_\theta} [Q_+(s_t, a)]$ and $A_+(t) := R_+(t) - b_+(s_t)$, so the surrogate objective for RePPO becomes

$$
\begin{aligned}
&\mathcal{L}_{\mathrm{RePPO}}(\theta) \\
&:= \mathbb{E} \left[ \min \left(r_\theta(t) A_+(t), \, \mathrm{clip}(r_\theta(t), 1-\varepsilon, 1+\varepsilon) A_+(t)\right) \right].
\end{aligned}
\tag{12}
$$

The algorithm is summarized in Alg. 1. Since RePPO differs from PPO solely through the use of a Q-critic and an EI-based advantage, its implementation is simple and the extra

---

[2]In Bayesian optimization, EI is the expected gain over the current best; here it is the improvement of an action over the best of $M-1$ other draws, a related but distinct notion.

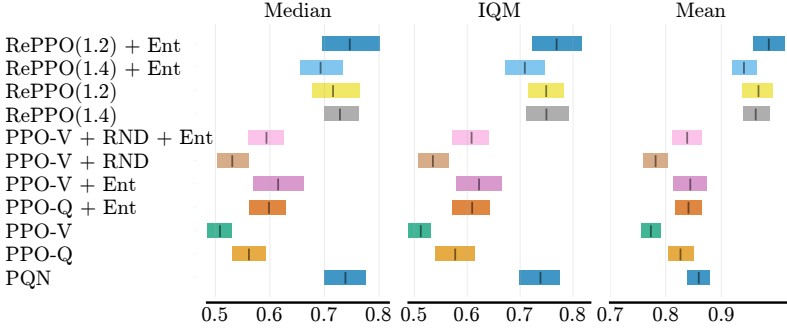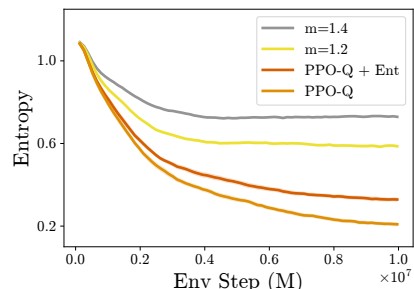

*Figure 3.* MinAtar results. **Left:** normalized scores aggregated with median, IQM, and mean across four games; boxes denote RLiable summaries over 10 seeds. RePPO, without entropy bonus, outperforms PPO-V, PPO-Q with entropy, and PPO-V + RND. **Right:** policy entropy during Breakout training. We observe that RePPO keeps high entropy without entropy bonus, indicating the promoted exploration.

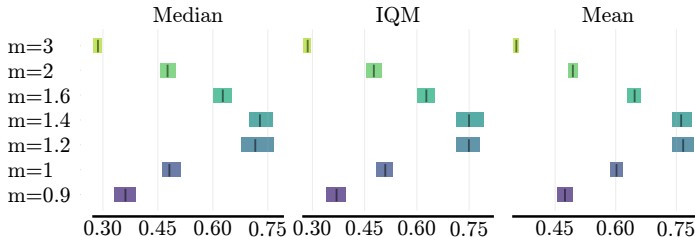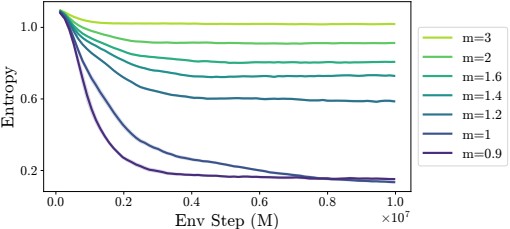

*Figure 4.* MinAtar results. **Effect of retry parameter** $m$. **Left:** median, IQM, and mean of normalized evaluation return across all games. The best performance occurred around $m \in [1.2, 1.4]$. **Right:** policy entropy on Breakout. Larger $m$ slowed entropy decay and encouraged exploration, while smaller $m$ leads to faster entropy decay.

computation is minimal (see App. E.2). We provide the advantage computation code in App. D.

**Q-replacement for efficient exploration.** When we compute the EI for the trajectory return, $R_+(s,a) := \mathrm{EI}_m(R(s,a); \pi_\theta, Q_\phi(s, \cdot))$, we expect $v_a = (R(s,a) - Q_\phi(s,a))_+$ to be zero in Eq. (10), since it measures an action's improvement over itself. In practice, an underspecified critic may yield $R(s,a) > Q_\phi(s,a)$, overestimating $R_+$ and causing the policy to overfit to $a$, harming exploration. To mitigate this, we *replace* the $a$-th element of $q = Q_\phi(s, \cdot)$ with $\mathcal{R}(s,a)$ when evaluating EI. This enforces $v_a = 0$ for the sampled action by construction and reduces spurious "self-improvement" caused by critic underestimation.

## 5. Experiments

To confirm the emergence of exploration in RePPO, we used three benchmark environments: *MinAtar* (Young & Tian, 2019), *Atari* (Bellemare et al., 2013), and *Craftax* (Matthews et al., 2024) (an open-ended, long-horizon exploration benchmark). MinAtar is a simplified version of Atari 2600 games providing Breakout, Asterix, Freeway, and Space Invaders; we used the `pgx` implementation (Koyamada et al., 2023) for efficient vectorized simulation. For Atari, we used 10 games from Bellemare's hard-exploration problems (Bellemare et al., 2016) to verify exploration bene-

fits (App. F). To further validate scalability, we used Craftax (Matthews et al., 2024), a vectorizable version of Crafter (Hafner, 2022), an open-ended RL environment.

### 5.1. MinAtar

Here, we evaluate RePPO on the MinAtar benchmark to demonstrate its effectiveness in promoting exploration. We also analyze the impact of key components, specifically the continuous retry parameter $m$ and the Q-replacement strategy, on the agent's performance and behavior.

**Baselines and hyperparameters.** We compared **RePPO** to **PPO-V** (state-value critic (Schulman et al., 2017)), **PPO-Q** (Q-critic), **PPO-V + RND** (Random Network Distillation (Burda et al., 2019)), and **PQN** (Gallici et al., 2025), a strong Q-learning baseline. PPO baselines followed the default `pgx` settings (based on PureJaxRL (Lu et al., 2022)), and PQN followed the official implementations. We report runs with and without entropy regularization (`+Ent` indicates it was enabled). Main results used RePPO with $m \in \{1.2, 1.4\}$; for speed comparisons vs. PPO-V and PPO-Q, see App. E. For RePPO, we tuned $m$ and set $\lambda = 0.8$ for the $\lambda$-return, keeping all other hyperparameters identical to PPO-V. PQN's official setup used 128 parallel environments; we used 1024 and adjusted the number of environments, minibatch size, and update epochs (tuning $\lambda$ and

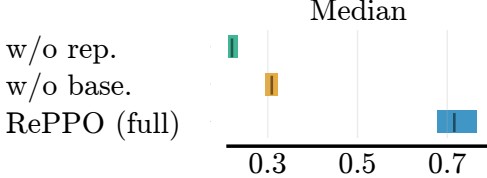

*Figure 5.* MinAtar results. Median (all games). The results show that either removing the action-independent baseline or the Q-replacement strategy substantially degrades performance.

the learning rate) to match the number of gradient updates. Additional comparisons with the unmodified PQN hyper-parameters (yielding ∼5× more updates) are provided in App. E.2. Full hyperparameters are listed in App. E.1.

**Training and evaluation.** Agents trained for 10M environment steps with 10 seeds; evaluation averaged 100 episodes/seed. For PPO variants, during evaluation, the action was selected by argmax over policy logits; during training, we sampled from the policy. We reported *normalized scores* across games (median, IQM, mean) via RLiable (Agarwal et al., 2021), normalized by the best returns across all methods; per-game scores are in App. E.2.

**Main results.** Figure 3 (Left) aggregates median, IQM, and mean across the four games. Among PPO variants, **RePPO** with $m \in [1.2, 1.4]$ performed best overall, even compared to entropy or RND bonuses. RePPO was comparable to PQN on median and IQM, and it consistently outperformed PQN on mean, indicating better tail performance. In Figure 3 (Right), RePPO *without* an entropy bonus maintained higher policy entropy than PPO *with* an entropy bonus. Finally, PPO-V + RND performed poorly when entropy was disabled, highlighting RND's reliance on entropy bonuses and that RePPO promotes exploration without bonuses.

**Effect of $m$: exploration–exploitation trade-off.** We tested $m \in \{0.9, 1.0, 1.2, 1.4, 1.6, 2, 3\}$ and reported median, IQM, and mean of normalized scores and policy entropy on Breakout (Figure 4). The sweep showed that increasing $m$ systematically slowed entropy decay, with returns peaking near $m \approx 1.2$–$1.4$ and falling when $m$ was larger or smaller, showing that the continuous parameter $m$ enabled more precise control of the exploration.

**Ablation on baseline and Q-replacement.** We further evaluated two key components of RePPO: the action-independent baseline and the Q-replacement strategy. With $m = 1.2$ without entropy bonus, we compared (i) *w/o base* (no baseline, Q-replacement enabled), (ii) *w/o rep* (baseline enabled, no Q-replacement), and (iii) *RePPO (full)* with both. Removing either component substantially degraded performance (Figure 5), indicating that both are necessary to realize the full benefits of RePPO (App. E.2).

*Table 1.* Craftax results. Mean and std of the % of the max reward (226) over 5 seeds. RePPO (1.2) is **bolded** and is comparable to the PPO with entropy and RND bonuses (underlined).

| Method | % of max (std) |
|---|---|
| PPO-V | 9.31(1.00) |
| PPO-V + Ent | 11.66(0.33) |
| RND | 9.62(1.54) |
| RND + Ent | 11.68(0.30) |
| PPO-Q | 9.31(0.93) |
| PPO-Q + Ent | 11.85(0.26) |
| **RePPO (1.2)** | **11.87(0.16)** |
| RePPO (1.4) | 10.79(0.19) |

### 5.2. Craftax

Craftax (Matthews et al., 2024) is an open-ended RL environment built on Crafter (Hafner, 2022) and NetHack (Küttler et al., 2020), in which the agent must both plan over long horizons and continually adapt to newly revealed parts of the environment. We use the symbolic version of Craftax.

**Setup.** We compared RePPO ($m \in \{1.2, 1.4\}$, no entropy bonus) against PPO-V, PPO-Q, and RND, each with/without an entropy bonus (coef. 0.01). All methods used the official Craftax implementation and hyperparameters[3]; RePPO changed only method-specific settings. Following Matthews et al. (2024), agents were trained for 1B timesteps with 5 seeds and evaluated for 100 episodes per seed; we reported mean and standard deviation across seeds as % of the max reward (226). Hyperparameters are in App. G.

**Results.** Table 1 shows that RePPO (1.2), without entropy or intrinsic bonuses, was competitive with PPO/RND variants that use such bonuses and outperformed PPO without them. App. G.1 shows RePPO maintained higher entropy even without bonuses (RND degraded without entropy), supporting that RePPO promotes exploration at larger scale.

## 6. Concluding Remarks

We conclude with the discussion and summary of the paper.

### 6.1. Scope and Future Work.

Our study focuses on discrete-action problems with a relatively small number of actions, where the full action-value vector is available and the ReMax gradient can be computed efficiently by sorting Q-values. Extending ReMax to large discrete or continuous action spaces via sampling-based estimators is an important direction. As discussed in Sec. 3, ReMax can be integrated into RL in multiple ways, and exploring alternatives to our PPO-based implementation, such

---

[3]https://github.com/MichaelTMatthews/Craftax_Baselines

as Q-learning (Mnih et al., 2013) and off-policy actor–critic methods (Haarnoja et al., 2018), is left for future work.

**Stochastic and deep Exploration.** One can distinguish between stochastic exploration and deep exploration (Gupta et al., 2018; Osband et al., 2016). The former injects randomness into action selection to diversify data collection, as in entropy-regularized methods (Haarnoja et al., 2018). The latter aims at temporally coherent information gathering, for example through visitation counts (Ostrovski et al., 2017) or posterior sampling (Osband et al., 2016), based on the mechanisms behind bandit algorithms with sublinear regret, such as UCB (Auer et al., 2002) and Thompson sampling (Thompson, 1933). In principle, both forms of exploration can emerge under ReMax. In the deterministic bandit example (Sec. 2.1), ReMax encourages the stochastic policy by slowing down the convergence, while in posterior bandits it leverages return uncertainty and yields empirically sublinear regret (Sec. 2.2). In our deep RL experiments, however, the observed behavior was more consistent with stochastic exploration, as indicated by increased policy entropy. We conjecture that more structured exploration may also have contributed to RePPO's performance gains, but confirming this requires more targeted empirical analysis. A promising direction is therefore to instantiate posterior-based ReMax by modeling epistemic uncertainty in Q-values explicitly, for example with ensembles or Bayesian critics. Finally, formal regret bounds for such variants would further strengthen the theoretical support for ReMax.

### 6.2. Summary.

We introduced **ReMax**, a novel RL objective that encourages exploration by *directly* maximizing reward, without auxiliary bonus terms. We demonstrated its effectiveness as an exploration mechanism in bandit settings and extended the formulation to RL. To optimize ReMax efficiently, we derived a new policy–gradient expression and relaxed the retry count to a continuous parameter $m$, enabling fine-grained control over the exploration–exploitation trade-off. We instantiated these ideas in **RePPO**, a PPO-style deep actor–critic method with a Q-critic, and showed on MinAtar that optimizing ReMax induces exploratory behavior and improves performance without entropy bonuses.

### Impact Statement

This work proposes a novel objective for exploration in RL. Its main potential impact is to improve exploration in domains where repeated attempts or diverse trials are important, including reasoning tasks with LLMs. Unlike exploration methods based on explicit bonuses or posterior sampling, ReMax does not require a separately designed exploration bonus or an explicit posterior model, but rather

is optimized using return samples. This may broaden the applicability of RL algorithms to settings where reliable bonus design or posterior modeling is difficult.

### Author Contributions

The project started in 2021 when Paavo Parmas proposed the idea to Sotetsu Koyamada who initially worked on it based on Paavo's guidance. Paavo moved to UTokyo and the project restarted in August 2024 when Soichiro Nishimori started working on it as his internship project under the guidance of Paavo Parmas. The contributions in the final manuscript are as follows:

Soichiro Nishimori: Lead writer, key implementations (the RePPO implementation and all experiments concerning it, as well as other final implementations) and experimentation, figures, discussion and literature search, proposal and experimentation with some early versions of continuous $m$ (not included in the final paper), and proposal of KL-extension to our earlier AC style approaches and its experiment (also not included in the final paper).

Paavo Parmas: Conceptualized the idea, did all of the key mathematical derivations (derivation of the gradient estimator, the expected improvement formulation, proposal of the final version of continuous $m$, came up with the bandit algorithm, proposed RePPO, hypothesized the adaptive properties of ReMax, e.g., the ones in Figure 1), example implementations (first REINFORCE implementation for ReMax, first ReMax bandit implementations, some debugging), significant comments and editing on the paper.

Sotetsu Koyamada: Initial development and implementation on the project. This led to a resetting based ReMax implementation similar to the A2C algorithm with experiments on MinAtar and maze domains. Wrote an early preprint on the project based on guidance mainly from Paavo.

Tadashi Kozuno was involved in some early discussion and guidance of Sotetsu. Toshinori Kitamura provided comments on the paper. Shin Ishii is the PI of the lab at Kyoto University where Paavo and Sotetsu belonged to when the project started. General supervision of Sotetsu at the time. Yutaka Matsuo is the PI of the UTokyo lab where Paavo belongs to. Funding acquisition and general management and supervision at the lab.

### Acknowledgment

This work was supported by JSPS KAKENHI Grant Number JP22H04998. SN was supported by JSPS KAKENHI Grant Number JP24KJ0818.

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

# Appendix Contents

**Reproducibility Statement.** We strive to make our results easy to reproduce. The ReMax objective, its closed-form components, and the policy-gradient estimator are specified in Secs. 3 and 4, with all assumptions and complete proofs in App. B. The exact advantage computation used in our implementation is provided in App. D. The MinAtar setup, environments, evaluation protocol, normalization, and all hyperparameters for every method, appear in Sec. 5 and App. E (including Tables and additional analyses). We report results over 10 random seeds and 100 evaluation episodes per seed and summarize performance with RLiable aggregates; per-game curves and ablations are in App. E.2. Complete details for the bandit experiments (problem setups, optimization procedure, and baselines) are in App. C. Official code is available at https://github.com/nissymori/remax-rl.

## A. Extended Related Work

This section provides a brief overview of exploration in RL.

### A.1. Exploration in RL

**Optimism in the Face of Uncertainty (OFU).** A prominent exploration strategy is OFU. Methods based on OFU mainly fall into two categories: confidence-based (Strehl & Littman, 2005; Jaksch et al., 2010; Dann & Brunskill, 2015) and bonus-based (Strehl et al., 2006; Strehl & Littman, 2008; Azar et al., 2017; Jin et al., 2018). While these approaches enjoy strong theoretical guarantees, they do not directly extend to deep RL, where explicit visitation counts are impractical to maintain. Bellemare et al. (2016) generalize visitation counts to enable OFU-style exploration in deep RL. There are several follow-ups that explore the count-based approach to deep RL, including Lobel et al. (2023); Ostrovski et al. (2017). In contrast to OFU, ReMax does not employ an explicit exploration mechanism; instead, exploration emerges by maximizing a multiple-retry objective defined over Q-values.

**Intrinsic Motivation (IM).** A complementary line of work that scales well with deep RL is intrinsic motivation (IM), which is broadly categorized into prediction-error-based, information-gain-based, and novelty-based methods.

Prediction-error-based methods learn a dynamics model and encourage visiting states (or state–action pairs) whose successor states are hard to predict (Stadie et al., 2015; Pathak et al., 2017); the idea dates back to Schmidhuber (1991b) and Thrun & Möller (1991). However, they can overemphasize inherently noisy states (Schmidhuber, 1991a). Information-gain-based methods instead seek states that reduce model uncertainty (Schmidhuber, 2010; Sun et al., 2011; Houthooft et al., 2016; Sukhija et al., 2025).

Novelty-based methods directly incentivize visiting "novel" states (or state–action pairs). Notions of novelty include pseudo-counts (Bellemare et al., 2016; Lobel et al., 2023; Taiga et al., 2020; Ostrovski et al., 2017), the estimated probability of a state appearing in a replay buffer (Fu et al., 2017), reachability-based metrics (Savinov et al., 2019), and intra-episode state diversity (Badia et al., 2020). Tang et al. (2017) discretize the state space with hashing to obtain counts.

These methods typically require estimating auxiliary density or dynamics models (e.g., transition models or visitation frequencies). By contrast, our method does not introduce any additional estimation targets. Two notable exceptions that avoid explicit density estimation are RND (Burda et al., 2019) and E-values (Fox et al., 2018). Combining such novelty detectors to flag unexplored states and to intensify the use of ReMax for exploration is an interesting future direction.

**Entropy Maximization.** Entropy-based exploration is widely used with policy-gradient and actor–critic methods (Williams & Peng, 1991; Mnih et al., 2016; Espeholt et al., 2018; Levine, 2018; Eysenbach & Levine, 2022). SAC is a popular example for continuous control (Haarnoja et al., 2018). While these approaches increase *action* entropy, increasing *state* entropy may align better with exploration goals (Pitis et al., 2020; Mutti et al., 2021), since action entropy alone promotes undirected exploration. Baram et al. (2021) propose maximizing transition entropy (the entropy of the next-state distribution given the current state) as a proxy for state entropy. Recent work explored beyond entropy regularization, such as f-divergence regularization (Labbi et al., 2026). In contrast, with ReMax, as shown in Sec. 2, directed (uncertainty-aware) exploration emerges from maximizing the retry objective.

**Posterior Sampling.** Another family draws inspiration from Thompson sampling (Thompson, 1933; Honda & Takemura, 2014; Agrawal & Goyal, 2017). Bootstrapped DQN (Osband et al., 2016) samples a Q-function from an ensemble to emulate Thompson sampling, later enhanced with randomized priors and value functions (Osband et al., 2018; 2019; Ishfaq et al., 2021). Other works use Bayesian neural networks to model the Q-function posterior (Azizzadenesheli et al., 2018; Bayrooti

et al., 2025; Osband et al., 2023) or adopt model-based posterior sampling (Sasso et al., 2023). Some works use Langevin Monte Carlo and efficient approximate sampling schemes to realize Thompson-style exploration in deep RL (Ishfaq et al., 2024; 2025). Integrating ReMax with such methods to *explicitly* model the Q-value distribution is a promising avenue.

## A.2. Retry-based objectives

Pass@K was introduced as an evaluation metric for LLM reasoning, measuring whether at least one of $K$ samples is correct (Chen et al., 2021). Beyond evaluation, recent work optimizes pass@K-like objectives directly. An early preprint of this submission proposed the ReMax objective and its simple optimization with REINFORCE already in 2022 (Koyamada et al., 2022), preceding subsequent max@K/pass@K policy-optimization works in LLM training. Walder & Karkhanis (2025) extend to continuous rewards via max@K, deriving unbiased estimators via reward transformations computed from mini-batches of sampled rewards/returns, and show that maximizing the best of $K$ trials improves both pass@K and pass@1. Their formulation coincides with ours when per-action values are fixed. Tang et al. (2025b) analyze $K$-sample objectives (pass@K, majority vote), studying bias–variance and KL efficiency and proposing leave-one-out control variates. Chen et al. (2025) propose RLVR with pass@K as the training signal, using bootstrap grouping and analytic advantages to cut rollout cost. These works consistently show that retry-based training improves exploration and robustness in reasoning tasks. Despite sharing a similar idea, several key distinctions separate the scope of our work from theirs. **Uncertainty over rewards:** We model uncertainty over rewards or Q-values and adapt exploration to (epistemic) uncertainty, whereas LLM reasoning benchmarks typically assume a fixed reward. **Retry feasibility:** In episodic RL, multiple returns from the same state are generally infeasible; we emulate retries via a learned Q-function and derive a policy gradient estimable from single-trajectory data. **Closed-form vs. sample-based computation:** Our ReMax objective and EI admit closed-form expressions in discrete-action RL that compute directly from the policy action probabilities (the $p$ / $\pi$ terms), whereas LLM pass@K/max@K training typically constructs objectives and gradient estimates from a batch of sampled completions and their rewards/returns (Walder & Karkhanis, 2025; Tang et al., 2025b). **Novel PG with continuous retry:** Moreover, prior work controls retries with an integer $K$, whereas we introduce a continuous retry parameter, enabling fine-grained exploration–exploitation trade-offs. This extension was enabled by our novel policy gradient formulation and the closed-form expression as shown in Sec. 4. Thus, although max@K coincides with ReMax under fixed values, our contribution extends retry-based objectives to uncertainty-aware, episodic RL with continuous control of retries. Recent work has proposed other objectives that move beyond standard expected return. Hamid et al. (2026) proposed polychromic objectives that explicitly encourage diverse behaviors by assigning high value to sets of rollouts only when they are both rewarding and diverse. Tang et al. (2025a) explored other aggregation functions for evaluation of the sequence of actions, such as max, min, and variance.

## A.3. Risk-sensitive RL

Risk-sensitive reinforcement learning modifies the criterion used to evaluate return distributions, rather than relying solely on their expectation. Classical risk-sensitive Markov decision processes study exponential utility or entropic criteria, which lead to risk-aware Bellman equations (Howard & Matheson, 1972; Fei et al., 2021). Another prominent criterion is Conditional Value-at-Risk (CVaR), which focuses on the lower tail of the return distribution and has been used to formulate safer RL objectives (Rockafellar et al., 2000; Ying et al., 2022). These approaches are related to our method in that they also go beyond the standard expected-return objective and reason about distributions of returns. However, their primary goal is to encode a risk preference over environmental returns. In contrast, ReMax evaluates the expected maximum over multiple sampled actions, thereby favoring actions that may yield high values under multiple retries and encouraging exploration.

Distributional RL provides another related perspective by explicitly modeling the full return distribution in deep RL (Bellemare et al., 2017; Dabney et al., 2018; Bellemare et al., 2023). This line of work represents the inherent randomness of returns, often viewed as *aleatoric* uncertainty, and has led to practical algorithms based on categorical (Bellemare et al., 2017) or quantile approximations of return distributions (Dabney et al., 2018; Bellemare et al., 2023). By contrast, the motivation of ReMax is primarily *epistemic*: it targets uncertainty arising from insufficient exploration and limited knowledge of action values. Nevertheless, ReMax can in principle be combined with distributional RL. For example, one could combine a retry-based objective with an explicit distributional value model, using the learned return distribution to evaluate the expected maximum over sampled action values or to control the degree of risk-seeking behavior. Exploring such combinations may be a promising direction.

# B. Proofs

This section contains the proofs of the propositions and theorems in the main text.

## B.1. Proof of Proposition 3.2

**Proposition 3.2.** Let $q = (q_1, \ldots, q_K)$ and write the order statistics $q_{(1)} \geq \cdots \geq q_{(K)}$, breaking ties arbitrarily, with aligned masses $\pi_{(j)}$. Define $C_0 := 0$ and $C_j := \sum_{u=1}^{j} \pi_{(u)}, j = 1, \ldots, K$. Then

$$J_{\text{ReMax}}^{M}(\pi, s, q) = q_{(1)} + \sum_{j=1}^{K-1} \left( q_{(j+1)} - q_{(j)} \right) \left( 1 - C_j \right)^{M}. \tag{13}$$

*Proof.* We compute the expectation by conditioning on the best sampled rank. Let $(j)$ denote the action with the $j$-th largest value after sorting the entries of $q$, with ties broken arbitrarily. For sampled actions $A_1, \ldots, A_M$, define

$$R^{\star} := \min\{j \in [K] : A_m = (j) \text{ for some } m \in [M]\}.$$

Then the maximum sampled value is

$$\max_{m \in [M]} q_{A_m} = q_{(R^{\star})}.$$

Therefore,

$$J_{\text{ReMax}}^{M}(\pi, s, q) = \mathbb{E}_{A_{[M]} \sim \pi} \left[ \max_{m \in [M]} q_{A_m} \right] \tag{14}$$

$$= \sum_{j=1}^{K} \mathbb{P}(R^{\star} = j) q_{(j)}. \tag{15}$$

It remains to compute $\mathbb{P}(R^{\star} = j)$. The event $R^{\star} = j$ occurs iff all $M$ draws miss the top-$(j-1)$ actions, but not all $M$ draws miss the top-$j$ actions. Since the total policy mass on the top-$j$ actions is $C_j$, we have

$$\mathbb{P}(R^{\star} = j) = \mathbb{P}(\text{all } M \text{ draws miss the top-}(j-1)) - \mathbb{P}(\text{all } M \text{ draws miss the top-}j) \tag{16}$$

$$= (1 - C_{j-1})^{M} - (1 - C_j)^{M}. \tag{17}$$

Let

$$\alpha_j := (1 - C_j)^{M}.$$

Then $\alpha_0 = 1$ and $\alpha_K = 0$, and hence

$$J_{\text{ReMax}}^{M}(\pi, s, q) = \sum_{j=1}^{K} (\alpha_{j-1} - \alpha_j) q_{(j)} \tag{18}$$

$$= \alpha_0 q_{(1)} + \sum_{j=1}^{K-1} \alpha_j \left( q_{(j+1)} - q_{(j)} \right) - \alpha_K q_{(K)} \tag{19}$$

$$= q_{(1)} + \sum_{j=1}^{K-1} \left( q_{(j+1)} - q_{(j)} \right) (1 - C_j)^{M}. \tag{20}$$

This completes the proof. $\square$

## B.2. Proof of Proposition 4.1.

**Proposition 4.1.** Let $W_{M-1} := \max\{q_{A_1}, \ldots, q_{A_{M-1}}\}$, we have:

$$\nabla_\theta J_{\text{ReMax}}^{M}(\theta, s, q) = M \mathbb{E}_{a \sim \pi_\theta} \left[ \nabla_\theta \log \pi_\theta(a|s) \mathbb{E}_{A_{[M-1]}} \left[ (q_a - W_{M-1})_+ \right] \right]. \tag{21}$$

*Proof.* We start by considering the PG with a per-term baseline.

$$\nabla_\theta J_M(\pi_\theta, s, q) = \mathbb{E}_{A_{[M]}} \left[ \sum_{m=1}^{M} \nabla_\theta \log \pi_\theta(A_m) \left( \max_{j \in [M]} q_{A_j} - b_m \right) \right]. \tag{22}$$

Choose $b_m$ as

$$b_m := W_{-m} := \max\{q_{A_1}, \ldots, q_{A_{m-1}}, q_{A_{m+1}}, \ldots, q_{A_M}\}, \tag{23}$$

which preserves unbiasedness since

$$\mathbb{E}_{A_m \sim \pi_\theta} \left[ \nabla_\theta \log \pi_\theta(A_m) \, b_m \right] = b_m \nabla_\theta \sum_{i=1}^{K} \pi_\theta(i) = 0, \tag{24}$$

With this baseline,

$$\max_{j \in [M]} q_{A_j} - W_{-m} = \left( q_{A_m} - W_{-m} \right)_+. \tag{25}$$

Hence

$$\nabla_\theta J_M(\pi_\theta, s, q) = \mathbb{E}_{A_{[M]}} \left[ \sum_{m=1}^{M} \nabla_\theta \log \pi_\theta(A_m)(q_{A_m} - W_{-m})_+ \right]. \tag{26}$$

**Condition on the first $M-1$ samples.** By symmetry of i.i.d. draws,

$$\nabla_\theta J_M(\pi_\theta, s, q) = M \, \mathbb{E}_{A_{[M-1]}} \left[ \mathbb{E}_{A_M \sim \pi_\theta} \left[ \nabla_\theta \log \pi_\theta(A_M) \left( q_{A_M} - W_{M-1} \right)_+ \right] \right], \tag{27}$$

where

$$W_{M-1} := \max\{q_{A_1}, \ldots, q_{A_{M-1}}\}. \tag{28}$$

For fixed $(q, A_{[M-1]})$, $W_{M-1}$ is a constant and $A_M \sim \pi_\theta$, so

$$\mathbb{E}_{A_M \sim \pi_\theta} \left[ \nabla_\theta \log \pi_\theta(A_M) \left( q_{A_M} - W_{M-1} \right)_+ \right] = \sum_{i=1}^{K} \pi_\theta(i) \, \nabla_\theta \log \pi_\theta(i) \, (q_i - W_{M-1})_+. \tag{29}$$

**Separate the action expectation.** Then, we have

$$\nabla_\theta J_M(\pi_\theta, s, q) = M \, \mathbb{E}_{i \sim \pi_\theta} \left[ \mathbb{E}_{A_{[M-1]} \sim \pi_\theta} \left[ \nabla_\theta \log \pi_\theta(i) \, (q_i - W_{M-1})_+ \right] \right], \tag{30}$$

which completes the proof. $\qquad\square$

### B.3. Proof of Proposition 4.3.

**Proposition 4.3.** Let $q \in \mathbb{R}^K$ be Q-values at a state, $\pi \in \Delta^{K-1}$ a policy, $R \in \mathbb{R}$ a reference, and $M \in \mathbb{N}$. Define $v_i := (R - q_i)_+$ and sort $q$ as $q_{(1)} \geq \cdots \geq q_{(K)}$ with aligned masses $\pi_{(j)}$. Define $C_0 := 0$ and $C_j := \sum_{u=1}^{j} \pi_{(u)}$, $j = 1, \ldots, K$. Then

$$\text{EI}_M(R; \pi, q) = v_{(1)} + \sum_{j=1}^{K-1} \left( v_{(j+1)} - v_{(j)} \right) (1 - C_j)^{M-1}. \tag{31}$$

*Proof.* By definition,

$$\text{EI}_M(R; \pi, q) = \mathbb{E}_{A_{[M-1]} \sim \pi} \left[ \left( R - \max_{m \in [M-1]} q_{A_m} \right)_+ \right].$$

Since $v_i = (R - q_i)_+$ and $q_{(1)} \geq \cdots \geq q_{(K)}$, we have $v_{(1)} \leq \cdots \leq v_{(K)}$ and

$$\left( R - \max_{m \in [M-1]} q_{A_m} \right)_+ = \min_{m \in [M-1]} v_{A_m}.$$

Using the same best-rank decomposition as in the proof of Proposition 3.2, with $M - 1$ draws, the probability that the best sampled rank is $j$ is

$$(1 - C_{j-1})^{M-1} - (1 - C_j)^{M-1}.$$

Therefore,

$$\mathrm{EI}_M(R; \pi, q) = \sum_{j=1}^{K} \left\{ (1 - C_{j-1})^{M-1} - (1 - C_j)^{M-1} \right\} v_{(j)}.$$

Applying the same telescoping argument as in Proposition 3.2 gives

$$\mathrm{EI}_M(R; \pi, q) = v_{(1)} + \sum_{j=1}^{K-1} \left( v_{(j+1)} - v_{(j)} \right)(1 - C_j)^{M-1}.$$

This completes the proof. $\qquad\square$

## C. Details of the Bandit Experiments

This appendix provides details of the bandit experiments.

### C.1. Binary Bandit

We can compute the ReMax objective for the binary bandit analytically as follows:

$$J^M_{\mathrm{ReMax}}(p) := 0.75 \cdot (1 - (1 - p)^M) + 0.25 \cdot (1 - p^M), \tag{32}$$

where $p$ is the probability of pulling arm 1.

### C.2. Bernoulli bandit

We describe (i) the Bernoulli-bandit setup, (ii) the experimental design for Fig. 1 (Center), and (iii) the computations for each method.

**(i) Bernoulli-bandit setup.** We consider two arms with rewards $R_i = \alpha_i X_i$, where $X_i \sim \mathrm{Bernoulli}(p_i)$ and $\mu_i = \mathbb{E}[R_i] = \alpha_i p_i$. For a fixed realization $r = (r_0, r_1)$ and a policy $\pi \in [0, 1]$ denoting $\Pr(a{=}0) = \pi$ (thus $\Pr(a{=}1) = 1 - \pi$), the ReMax objective with $M{=}2$ i.i.d. draws is

$$J^2(\pi \mid r) = \pi^2 r_0 + 2\pi(1 - \pi) \max\{r_0, r_1\} + (1 - \pi)^2 r_1.$$

Taking expectation over Bernoulli outcomes $e_1 = (\alpha_0, 0)$, $e_2 = (0, \alpha_1)$, $e_3 = (\alpha_0, \alpha_1)$, $e_4 = (0, 0)$ with probabilities $p_{e_1} = p_0(1 - p_1)$, $p_{e_2} = (1 - p_0)p_1$, $p_{e_3} = p_0 p_1$, $p_{e_4} = (1 - p_0)(1 - p_1)$ gives

$$\mathbb{E}\left[J^2(\pi \mid R)\right] = \sum_{k=1}^{4} p_{e_k} J^2(\pi \mid e_k).$$

**(ii) Experimental design.** We fix $p_0 = 1$ and $\alpha_0 = 2$, and sweep arm-1 scale $\alpha_1 \in [1, 10]$ while adjusting $p_1$ to keep the mean constant: $\alpha_1 p_1 = 1$. For each $\alpha_1$, we evaluate $\pi^\star(a{=}1) = 1 - \pi^\star$ under $M = 2$ and plot it together with the softmax baseline in Fig. 1 (Center).

**(iii) Method computations.** *ReMax ($M{=}2$).* Compute $\mathbb{E}\left[J^2(\pi \mid R)\right]$ via the above event decomposition and maximize over $\pi \in [0, 1]$ (we use a dense numerical search; a closed form exists because the objective is quadratic in $\pi$). *Softmax (entropy-regularized).* For temperature $\beta > 0$ (we use $\beta{=}1$),

$$\pi_{\mathrm{soft}}(a{=}1) = \frac{\exp(\mu_1/\beta)}{\exp(\mu_0/\beta) + \exp(\mu_1/\beta)}, \quad \mu_0 = \alpha_0 p_0, \ \mu_1 = \alpha_1 p_1 \ (= 1 \text{ in our sweep}).$$

## C.3. Bandit with Posterior

---
**Algorithm 2** ReMax with Posterior
---
1: Initialize prior $\Pi_{0,i}$ for each arm.
2: **for** $t = 1, 2, \ldots, T$ **do**
3:     Compute $\pi_t = \arg\max_\pi \mathbb{E}_{\mu_t \sim \Pi_t} \left[ J^M_{\text{ReMax}}(\pi; \mu_t) \right]$ by Alg. 3.
4:     Play $a_t \sim \pi_t$, observe $r_t \in \mathbb{R}$.
5:     Update posterior of arm $a_t$, $\Pi_{t+1,a_t}$, by the posterior update rule.
6: **end for**

---

---
**Algorithm 3** ReMax optimization
---
**Require:** Posterior $\Pi_t$, batch size $B$, draws $M$, epochs $S$ and policy $\theta_{t-1}$
1: Initialize policy $\pi_\theta$ with $\theta_{t-1}$
2: **for** $s = 1, 2, \ldots, S$ **do**
3:     Sample $\mu^{b,t} \sim \Pi_t$ for $b = 1, 2, \ldots, B$.
4:     Compute $J^M_{\text{ReMax}}(\pi_\theta; \mu^{b,t})$ for $b = 1, 2, \ldots, B$.
5:     Update $\pi_\theta$ by gradient ascent.
6: **end for**
7: **return** $\pi_t = \pi_\theta$
---

To confirm empirically sublinear regret, we conduct a posterior-driven bandit experiment in two settings: Beta–Bernoulli and Gaussian–Gaussian. We consider a ground-truth prior $\Pi^*$ over the arm means $\{\mu_i\}_{i=1}^K$. Initially, each $\mu_i \sim \Pi^*$ and the learner's prior is $\Pi_0 = \Pi^*$. At each round $t$, we select $a_t$, observe $r_t$ drawn with mean $\mu_{a_t}$, and update the posterior $\Pi_{t+1,a_t}$.

**Beta–Bernoulli.**    The prior $\Pi^*$ is $\text{Beta}(\alpha_0, \beta_0)$ with $\alpha_0 = \beta_0 = 1$ for all arms. Rewards are Bernoulli with mean $\mu_i$. The posterior update is

$$\Pi_{t+1,i} = \text{Beta}(\alpha_t + r_t, \, \beta_t + 1 - r_t).$$

**Gaussian–Gaussian.**    The prior $\Pi^*$ is $\mathcal{N}(\mu_0, \sigma_0^2)$ with $\mu_0 = 0$ and $\sigma_0^2 = 1$ for all arms. Rewards are $\mathcal{N}(\mu_i, \sigma_R^2)$ with $\sigma_R^2 = 1$. The posterior update is

$$\Pi_{t+1,i} = \mathcal{N}(\mu_{i,t+1}, \sigma_{i,t+1}^2), \qquad \sigma_{i,t+1}^2 = \left( \tfrac{1}{\sigma_{i,t}^2} + \tfrac{1}{\sigma_R^2} \right)^{-1}, \quad \mu_{i,t+1} = \sigma_{i,t+1}^2 \left( \tfrac{\mu_{i,t}}{\sigma_{i,t}^2} + \tfrac{r_t}{\sigma_R^2} \right).$$

**ReMax optimization.**    Arm selection by ReMax is shown in Alg. 2. At each $t$, we optimize the ReMax objective via Alg. 3, the exact computation of the objective from Prop. 3.2 with batch size $B = 16$ and epochs $S = 50$. For ease of optimization, at each round $t$ we initialize the policy $\theta_t$ with the parameters from the previous round $t - 1$. To avoid optimizer state carrying over across rounds, we reinitialize the optimizer at each round.

**Baselines.**    We compare against Thompson sampling (Thompson, 1933; Honda & Takemura, 2014; Agrawal & Goyal, 2017) and UCB (Auer et al., 2002), both with sublinear-regret guarantees. Thompson sampling: sample $\mu_a$ from $\Pi_{t,a}$ and select $a = \arg\max_a \mu_a$. UCB: after initializing by pulling all arms, select the arm with highest empirical mean plus a bonus $c\sqrt{\log(t)/(2N_a)}$, where $N_a$ is the number of pulls; we use $c = 1.0$ for both settings. To compare with the entropy-regularized exploration, we prepared a Softmax baseline, where we took the softmax of the posterior means of each arm and selected the arm following the softmax distribution. We fine-tune the temperature parameter in $(0.01, 0.1, 1.0)$ and used 0.1 for the experiments.

## D. RePPO

Below is the code for the advantage computation in RePPO.

*Listing 1.* Advantage Computation for RePPO

```
def expected_improvement_min(
    R: jnp.ndarray,  # (B, N_ref)
    q: jnp.ndarray,  # (B, K)
    pi: jnp.ndarray, # (B, K)
    M: float,
):
    """EI_M(R;pi) = E[min_{1..M} (R - q_A)_+],  A~pi i.i.d.
       Returns: (B, N_ref)
    """
    idx = jnp.argsort(-q, axis=-1)
    q_sorted = jnp.take_along_axis(q, idx, axis=-1)
    pi_sorted = jnp.take_along_axis(pi, idx, axis=-1)

    C = jnp.cumsum(pi_sorted, axis=-1)   # (B, K)

    v = jnp.maximum(R[..., None] - q_sorted[:, None, :], 0.0)  # (B, N_ref, K)
    v_first = v[..., 0]
    dv = v[..., 1:] - v[..., :-1]
    eps = 1e-8
    w = jnp.power(jnp.clip(1.0 - C[..., :-1], eps, 1.0), M)  # (B, K-1)
    EI = v_first + jnp.sum(dv * w[:, None, :], axis=-1)       # (B, N_ref)
    return EI

def reppo_advantage(
    R: jnp.ndarray,        # (B,) or (B,1)
    q: jnp.ndarray,        # (B, K)
    pi: jnp.ndarray,       # (B, K)
    action: jnp.ndarray,   # (B,)
    M: float,
):
    """Compute RePPO advantage with Q-replacement by return.
       Returns: (B,)
    """
    if R.ndim == 1:
        R = R[:, None]  # (B, 1)

    # Q-replacement: set q[action] <- R
    q_ref = q.at[jnp.arange(q.shape[0]), action].set(R[:, 0])

    # Improvement terms
    R_plus = expected_improvement_min(R, q_ref, pi, M - 1)[..., 0]  # (B,)
    q_plus = expected_improvement_min(q, q_ref, pi, M - 1)          # (B, K)
    baseline = jnp.sum(pi * jax.lax.stop_gradient(q_plus), axis=-1)  # (B,)
    return R_plus - baseline
```

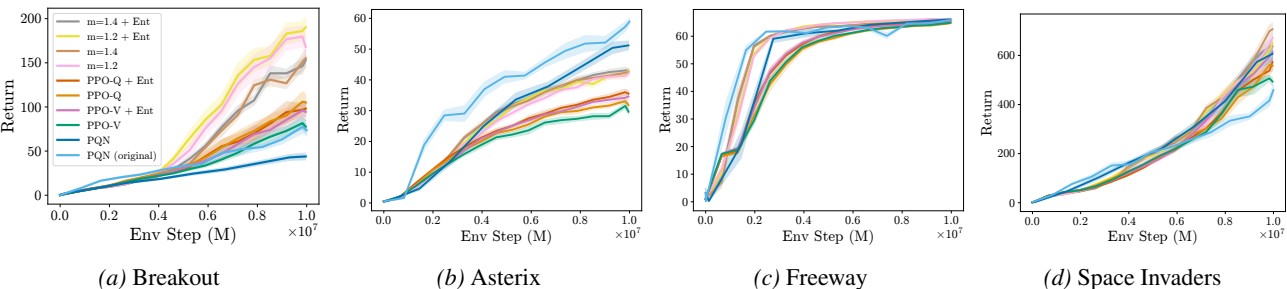

*Figure 6.* Per-game learning curves, ordered as Breakout, Asterix, Freeway, and Space Invaders. Mean ± s.e. over 10 seeds.

## E. MinAtar Experiment

In this appendix, we provide additional details on the MinAtar experiment.

### E.1. Experimental setup

**Network architecture.** We use the same network architecture as the public implementations of PPO and PQN (official implementation). **PPO (Actor–Critic).** A shared CNN (Conv 2×2 + ReLU + avg-pool) and MLP produce a latent state, which branches into (i) an actor head with two hidden layers (ReLU/Tanh) outputting action logits, and (ii) a critic head with two hidden layers outputting per-action Q-values (i.e., a Q-critic instead of a scalar $V$). **PQN (Q-Network).** Inputs are scaled (optional BatchNorm), then passed to a CNN feature extractor (Conv 3×3 with LayerNorm/BatchNorm + ReLU) and an MLP; a final linear layer outputs Q-values for all actions. This is a single-head, value-based model tailored to pure Q-learning.

**Code references.** We list the code references used in our experiments.

- MinAtar: `pgx` implementation (Koyamada et al., 2023)[4]

- PPO: `pgx` implementation (Koyamada et al., 2023)[5], which is also based on `purejaxrl` (Lu et al., 2022)[6]

- PQN: official implementation (Gallici et al., 2025)[7]

**Hyperparameters.** For PQN with 1024 parallel environments, we tuned the learning rate (0.0005, 0.001) and GAE $\lambda$ (0.65, 0.8, 0.95) for each environment. For RND, we tuned the learning rate (0.001, 0.0003, 0.0001) of the RND network and bonus coefficient (0.5, 1.0, 1.5) for each environment. We report the hyperparameters for RePPO, PPO-V, PPO-Q, and PQN (both original and tuned configurations) in Tables 2, 3, and 4 (App. J), respectively.

**Training and evaluation.** Agents are trained for 10M environment steps with 10 random seeds. During evaluation, we average 100 test episodes per seed. For PPO variants, evaluation uses the argmax of the policy's logits, whereas training samples actions from the policy. We report *normalized scores* aggregated across games using median, interquartile mean (IQM), and mean, following the RLiable framework (Agarwal et al., 2021). Scores are normalized by the maximum score achieved across all methods. Per-game scores are reported in App. E.2. The maximum scores used for normalization are: Breakout 251.15, Asterix 64.95, Freeway 67.05, Space Invaders 880.91.

### E.2. Additional results

We present supplementary analyses to complement the main results. Unless stated otherwise, curves show the mean and standard error over 10 seeds.

---

[4]https://github.com/sotetsuk/pgx

[5]https://github.com/sotetsuk/pgx

[6]https://github.com/luchris429/purejaxrl

[7]https://github.com/mttga/purejaxql

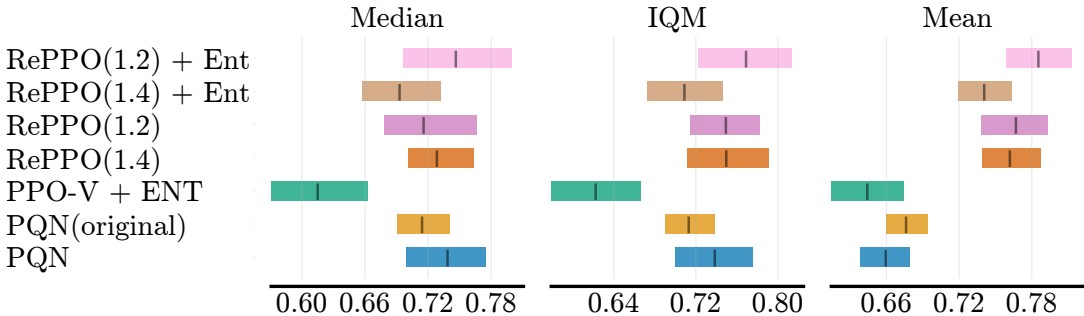

*Figure 7.* Aggregate metrics (Median, IQM, Mean) across all games.

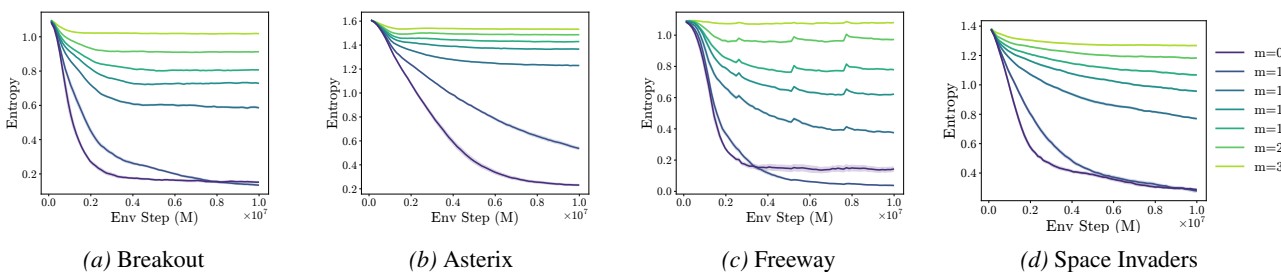

*(a)* Breakout       *(b)* Asterix       *(c)* Freeway       *(d)* Space Invaders

*Figure 8.* Policy entropy for all environments over $m \in \{0.9, 1.0, 1.2, 1.4, 1.6, 2, 3\}$, ordered as Breakout, Asterix, Freeway, and Space Invaders.

**Per-game results.** Fig. 6 reports learning curves for each MinAtar game. RePPO clearly outperforms PPO on *Breakout* and *Asterix*, and converges faster than the baselines on *Freeway*, consistent with the main findings. On *Asterix*, PQN outperforms all other methods.

**Entropy across environments.** Fig. 8 shows policy entropy for all environments and $m \in \{0.9, 1.0, 1.2, 1.4, 1.6, 2, 3\}$. As expected, entropy aligns with the retry parameter $m$: small $m$ leads to rapid entropy decay, while larger $m$ sustains higher entropy. Entropy can also be tuned more finely by varying $m$ between these values.

**Comparison to the original PQN.** In the main text, we compared RePPO to PQN tuned for our setup. Fig. 7 adds results for the original PQN. Overall, the original PQN is comparable to the one that we tuned for our setup. PQN is comparable to RePPO on Median and IQM, but underperforms on Mean, the same trend we observed in the main text. This confirms that the difference of the setup does not affect the performance of PQN that much, confirming the validity of our analysis.

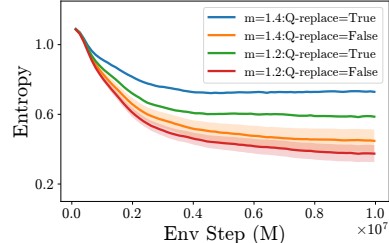

*Figure 9.* Entropy with and without Q-replacement on *Breakout*.

**Details on the analysis of Q-replacement.** Recall that Q-replacement substitutes the critic estimate $Q(s, a)$ at the sampled action with the empirical return $R$ before computing the EI advantage. We hypothesize that this substitution mitigates distortions of the EI advantage caused by inaccurate Q-values, which may otherwise reduce exploration. To test this, we compare entropy with and without Q-replacement on *Breakout* under the same retry parameters $m \in \{1.2, 1.4\}$ (Fig. 9). Across both values of $m$, the no-replacement variant exhibits consistently lower entropy throughout training. This suggests that Q-replacement helps preserve the exploratory pressure induced by the retry parameter and contributes to RePPO's overall performance.

**The standard deviation of the EI-based advantage.** Fig. 10 reports the standard deviation of the EI-based advantage for *Breakout* during training, computed over the minibatch and averaged over 5 random seeds, with error bars denoting the standard error across seeds. This statistic measures the variability of the scalar advantage signal used in the actor update,

and can therefore serve as a diagnostic of the stability of the advantage estimates.

We observe that retry parameters that performed well in our experiments, such as $m = 1.2$ and $m = 1.4$, tend to yield smaller advantage standard deviations than a less exploratory setting such as $m = 0.9$. This suggests that the same range of $m$ that promotes exploration can also produce a more stable EI-based advantage signal on *Breakout*. We emphasize, however, that this observation does not by itself isolate the cause of the variance reduction; it may reflect the combined effects of the EI transformation, the induced policy entropy, critic accuracy, and Q-replacement.

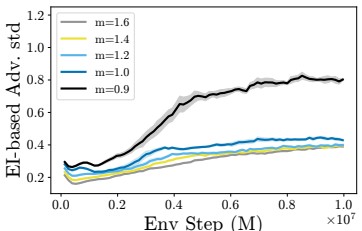

*Figure 10.* Standard deviation of the EI-based advantage on *Breakout*.

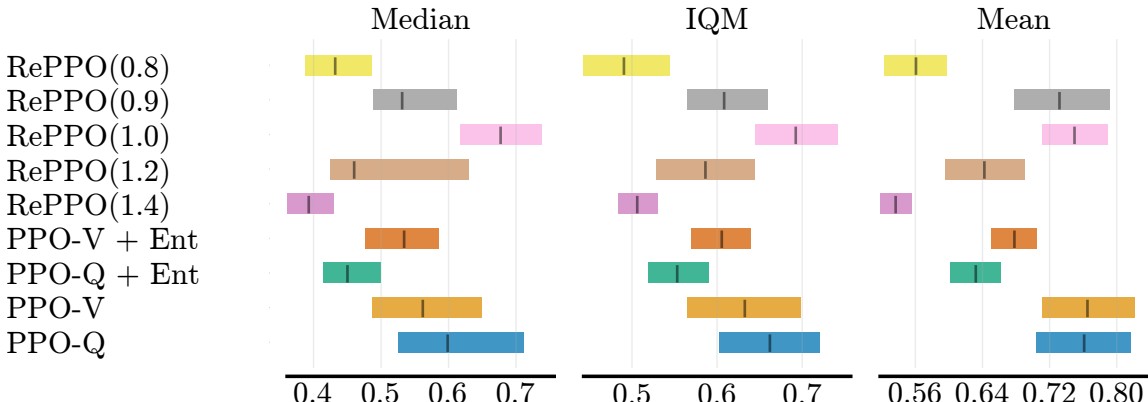

*Figure 11.* Normalized scores aggregated with median, IQM, and mean across 10 games; boxes denote RLiable summaries over 5 seeds. For RePPO, we see the performance peak around $m = 0.9$ to $1.0$, and PPO without entropy performs better than that with entropy. This indicates that those environments indeed require less exploration.

## F. Atari Experiment

We use the 10 Atari environments identified as dense-reward, hard-exploration problems by Bellemare et al. (2016), namely: Alien, Amidar, BattleZone, Frostbite, Hero, Ms. Pacman, Q*bert, Surround, Wizard of Wor, and Zaxxon. Our implementation is based on the CleanRL codebase (Huang et al., 2022) and employs EnvPool (Weng et al., 2022) for parallel environment execution.

### F.1. Experimental setup

We follow the hyperparameters listed in Tables 5 and 6 (App. J), which are adapted from CleanRL. As baselines, we use PPO-V and PPO-Q, each evaluated with and without entropy regularization. RePPO is trained with retry parameters $m = 0.8, 0.9, 1.0, 1.2, 1.4$, and we report results for all configurations. Training is conducted for $1 \times 10^7$ environment steps. Evaluation is carried out in parallel with training using eight environments, and we report normalized scores at the final training step. All results are averaged over five random seeds and aggregated across the 10 environments using the RLiable framework (Agarwal et al., 2021) to compute the median, interquartile mean (IQM), and mean performance and it is shown in Fig. 11. We also plotted the raw return curves for all games in Fig. 12.

### F.2. Results

Fig. 11 shows the normalized scores aggregated with median, IQM, and mean across 10 games. Across all environments, methods with weaker exploration, such as PPO-V, PPO-Q, and RePPO with $m = 0.9$ or $1.0$, achieve the highest performance. In contrast, RePPO with larger retry parameters ($m = 1.2$ and $1.4$) and entropy-regularized PPO variants exhibit lower performance. This results in a performance peak around $m = 0.9$ to $1.0$. These findings suggest that, in practice, Bellemare's suite of hard-exploration environments may require relatively little exploration. Fig. 13 shows the evolution of policy entropy during training on all games. For $m = 1.2$ and $1.4$, the policy maintains higher entropy, indicating that RePPO indeed encourages more exploratory behavior even in complex, pixel-based environments such as Atari. Taken together, these results demonstrate that RePPO promotes exploration in Atari, but also remains flexible: when little exploration is needed, choosing a smaller retry parameter (e.g., $m = 0.9$ or $1.0$) yields strong performance. In contrast, larger values encourage exploration when desired.

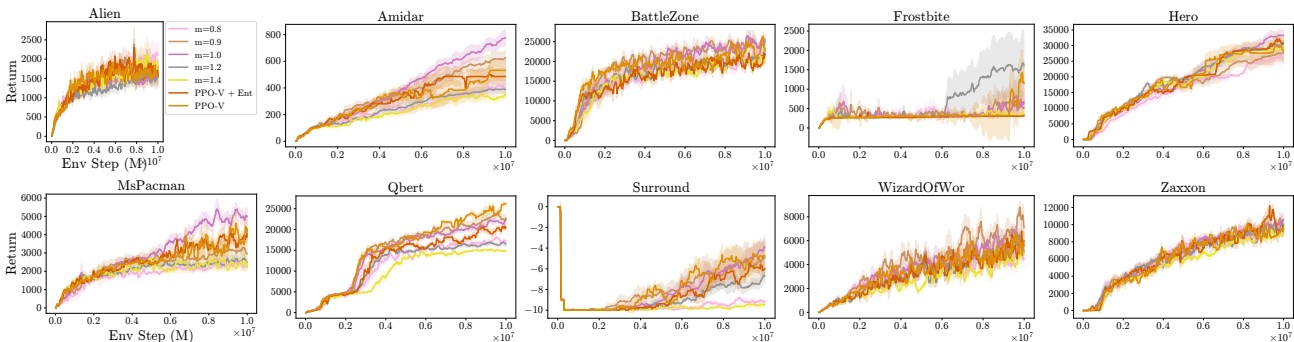

*Figure 12.* Plot of the return curves for all games. Mean $\pm$ s.e. over 5 seeds. Overall, PPO without entropy and RePPO with $m = 0.9$ and 1.0 perform better than that with entropy and RePPO with $m = 1.2$ and 1.4. This indicates that those environments indeed require less exploration.

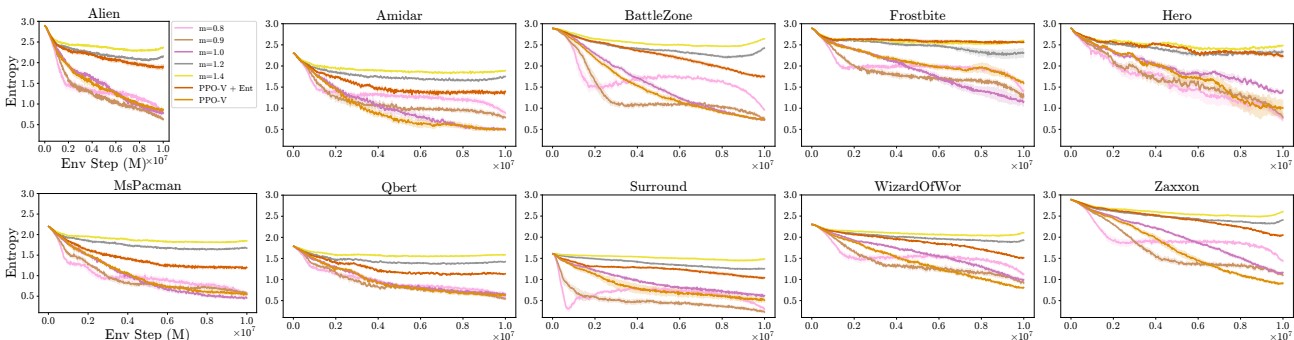

*Figure 13.* Policy entropy during training on all games. Mean $\pm$ s.e. over 5 seeds. RePPO with $m = 1.2$ and 1.4 maintains higher entropy, while that with $m = 0.9$ and 1.0 exhibits faster entropy decay, demonstrating the RePPO's ability to control the trade-off between exploration and exploitation.

## G. Craftax Experiment

**Hyperparameters.** For PPO-V and PPO-V + RND (RND), we used the same hyperparameters as in the original implementation[8]. For RePPO, we adopted the same hyperparameters and modified only the RePPO-specific retry parameter $m$. The full set of hyperparameters is listed in Table 7 (App. J).

### G.1. Results

Fig. 14 reports the policy entropy during training over 1B environment steps for RePPO ($m \in \{1.2, 1.4\}$, without entropy bonus) and the baseline PPO variants with and without entropy regularization, as well as PPO-V combined with an RND (Burda et al., 2019) intrinsic bonus. Three regimes are clearly visible. First, both RePPO configurations maintain the highest entropy throughout training, with $m = 1.4$ sustaining a noticeably higher level than $m = 1.2$, mirroring the behavior we observed on MinAtar and Atari. Second, PPO-V and PPO-Q without an entropy bonus collapse to low-entropy policies within the first $\sim$10% of training, indicating premature exploitation. Third, the entropy-

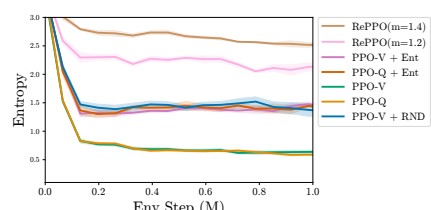

*Figure 14.* Entropy (Craftax).

regularized baselines (PPO-V/Q + Ent) and PPO-V + RND settle at an intermediate entropy level controlled by the bonus coefficient. Crucially, RePPO achieves this elevated entropy purely through its retry mechanism, without any explicit exploration bonus. Combined with Table 1 in Sec. 5, where RePPO (1.2) matches the performance of entropy-regularized PPO and PPO + RND and clearly outperforms PPO variants without an entropy bonus, this confirms that RePPO effectively promotes exploration even on large-scale, open-ended environments such as Craftax—without requiring entropy

---

[8]https://github.com/MichaelTMatthews/Craftax_Baselines

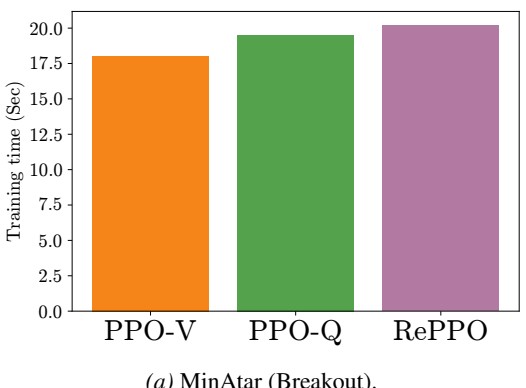

*(a)* MinAtar (Breakout).

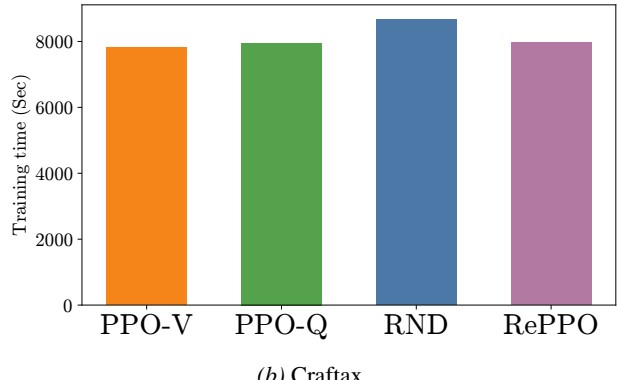

*(b)* Craftax.

*Figure 15.* Average training time of RePPO and baseline methods.

regularization or an additional intrinsic-reward model.

## H. Speed Benchmark

**MinAtar.** We compare the training speed of RePPO, PPO-V, and PPO-Q in MinAtar. Fig. 15a reports the average wall-clock time on *Breakout* for 10M timesteps (no evaluation), averaged over 5 seeds. Hyperparameters match Sec. 5. Since our implementation uses the JAX framework, we exclude JIT warmup time for all methods. As expected, RePPO is slower than PPO-V and PPO-Q because it computes EI for the advantage. However, the additional cost is comparable to the gap between PPO-V and PPO-Q, that is, to the overhead incurred by replacing a $V$-critic with a $Q$-critic. This suggests that the additional cost of RePPO, such as sorting Q-values and computing EI, is negligible compared to the improvement in performance.

**Craftax.** We also compare the training speed of RePPO with PPO-V and PPO-V + RND in Fig. 15b. Computation time is averaged over 5 random seeds. RePPO performs comparably to PPO-V and PPO-Q, and runs faster than PPO-V + RND—an expected outcome, as RePPO does not require additional models beyond those used in PPO-V and PPO-Q.

## I. LLM Usage

We made limited, assistive use of Large Language Models (LLMs) for presentation-related tasks. In particular, we used LLMs to revise wording for readability, provide minor assistance when organizing proof steps, suggest code refactoring options, and propose small figure improvements. LLMs were not used for research ideation, study design, or the development of substantive scientific contributions.

# J. Hyperparameter Tables

This appendix collects the hyperparameter tables referenced throughout the experiment sections.

*Table 2.* Hyperparameters for RePPO (MinAtar).

| Name (symbol) | Value |
| --- | --- |
| total timesteps | $1.0 \times 10^7$ |
| learning rate | $3.0 \times 10^{-4}$ |
| rollout length | 128 |
| parallel envs | 1024 |
| update epochs | 3 |
| minibatch size | 1024 |
| discount $\gamma$ | 0.99 |
| GAE $\lambda$ | 0.8 |
| clip $\varepsilon$ | 0.2 |
| entropy coefficient | (0.0, 0.01) |
| value loss coefficient | 0.5 |
| max grad norm | 0.5 |
| optimizer | Adam (with global-norm clip) |
| RePPO draws $m$ | (1.2, 1.4) |

*Table 3.* Hyperparameters for PPO-V and PPO-Q (MinAtar).

| Name (symbol) | Value |
| --- | --- |
| total timesteps | $1.0 \times 10^7$ |
| learning rate | $3.0 \times 10^{-4}$ |
| rollout length | 128 |
| parallel envs | 1024 |
| update epochs | 3 |
| minibatch size | 1024 |
| discount $\gamma$ | 0.99 |
| GAE $\lambda$ | 0.95 |
| clip $\varepsilon$ | 0.2 |
| entropy coefficient | (0.0, 0.01) |
| value loss coefficient | 0.5 |
| max grad norm | 0.5 |
| optimizer | Adam (with global-norm clip) |
| **RND related** | |
| learning rate | 0.001 (Asterix), 0.0003 (Breakout, Freeway), 0.0001 (Space Invaders) |
| bonus coefficient | 1.5 (Freeway), 1.0 (others) |

*Table 4.* Hyperparameters for PQN on MinAtar: original (128 parallel envs) and tuned (1024 parallel envs).

| Name | Original | Tuned (1024 envs) |
|------|----------|-------------------|
| total timesteps | $1.0 \times 10^7$ | $1.0 \times 10^7$ |
| parallel envs | 128 | 1024 |
| rollout length | 32 | 128 |
| minibatch size | 128 | 1024 |
| update epochs | 2 | 3 |
| learning rate | $5.0 \times 10^{-4}$ | $1.0 \times 10^{-3}$ |
| GAE $\lambda$ | 0.65 | 0.8 |
| LR linear decay | True | True |
| max grad norm | 10.0 | 10.0 |
| discount $\gamma$ | 0.99 | 0.99 |
| $\varepsilon$-start / finish / decay ratio | 1.0 / 0.05 / 0.1 | 1.0 / 0.05 / 0.1 |
| normalization type | layer_norm | layer_norm |
| optimizer | RAdam (global-norm clip) | RAdam (global-norm clip) |

*Table 5.* Hyperparameters for RePPO (Atari).

| Name (symbol) | Value |
|---------------|-------|
| total timesteps | $1.0 \times 10^7$ |
| learning rate | $2.5 \times 10^{-4}$ |
| rollout length | 128 |
| parallel envs | 128 |
| update epochs | 4 |
| minibatch size | 128 |
| discount $\gamma$ | 0.99 |
| GAE $\lambda$ | 0.95 |
| clip $\varepsilon$ | 0.1 |
| entropy coefficient | (0.0, 0.01) |
| value loss coefficient | 0.5 |
| max grad norm | 0.5 |
| optimizer | Adam (with global-norm clip) |
| RePPO draws $m$ | (0.8, 0.9, 1.0, 1.2, 1.4) |

*Table 6.* Hyperparameters for PPO-V and PPO-Q (Atari).

| Name (symbol) | Value |
|---------------|-------|
| total timesteps | $1.0 \times 10^7$ |
| learning rate | $2.5 \times 10^{-4}$ |
| rollout length | 128 |
| parallel envs | 128 |
| update epochs | 4 |
| minibatch size | 128 |
| discount $\gamma$ | 0.99 |
| GAE $\lambda$ | 0.95 |
| clip $\varepsilon$ | 0.1 |
| entropy coefficient | (0.0, 0.01) |
| value loss coefficient | 0.5 |
| max grad norm | 0.5 |
| optimizer | Adam (with global-norm clip) |

*Table 7.* Hyperparameters for RePPO, PPO-V and PPO-Q (Craftax).

| Name (symbol) | Value |
|---|---|
| total timesteps | $1.0 \times 10^9$ (1B) |
| learning rate | $2 \times 10^{-4}$ |
| rollout length | 64 |
| parallel envs | 1024 |
| update epochs | 4 |
| minibatch size | 8192 |
| discount $\gamma$ | 0.99 |
| GAE $\lambda$ | 0.8 |
| clip $\varepsilon$ | 0.2 |
| entropy coefficient | {0.0, 0.01 (for PPO-V and PPO-Q)} |
| value loss coefficient | 0.5 |
| max grad norm | 1.0 |
| **RND related** | |
| learning rate | $3.0 \times 10^{-4}$ |
| bonus coefficient | 1.0 |
| **RePPO related** | |
| RePPO draws $m$ | (1.2, 1.4) |

