# OpenReview forum: "Emergence of Exploration in Policy Gradient Reinforcement Learning via Retrying"
_ICML.cc/2026/Conference — ICML 2026 regular_

### Official Review · Reviewer_wtaF · 2026-03-09

**Soundness:** 3
**Presentation:** 2
**Significance:** 2
**Originality:** 3
**Overall Recommendation:** 4
**Confidence:** 2

**Summary:**

This paper uses the concept of retrying to modify the original RL objective to one that allows finer control over exploration with a continuous parameter. The remax operator considers an expectation over the maximum of $M$ samples, and so, this gives a type of optimistic objective which effectively selects the best realization when $M>1$. They illustrate examples over the basic bandit setting to show that it helps in exploration in this case. Next they proceed to the MDP setting, where they adapt the policy gradient update so that it can handle single trajectory returns with a closed form expression. They then combine this with the PPO algorithm to give the RePPO algorithm, which is used to perform experiments in 3 main benchmarks: MinAtar, Atari and Craftar. The results are much more clear in MinAtar, where performance metrics and ablations clearly show trends which support the usefulness of the remax approach. However, in Craftar and Atari the overall results are slightly less convincing, but still seem to be in a positive direction in support of remax.

**Compliance With Llm Reviewing Policy:**

Affirmed.

**Final Justification:**

I'm inclined to agree with the point 1 in weaknesses made by reviewer Zv2H, although I think this is challenging to show and so I maintain that the current results are satisfactory. Due to the minor limitations in the experiments that I mentioned in my review and rebuttal response, but overall satisfactory outcome, I would like to maintain my current mild positive score.

**Key Questions For Authors:**

Please see my main comments: 2,5,6,7

**Limitations:**

Yes

**Strengths And Weaknesses:**

The theoretical results of this paper seem to be correct and the overall idea makes sense. Some minor aspect about the presentation could be improved. I have mentioned some of these points below. The overall empirical results are convincing to an extent, but it does not seem completely obvious that using the remax approach is helping with exploration. In my opinion the overall results are positive but mild. The ideas seem fairly original, but I do think they are quite similar to the ideas of risk-aware RL: which have not been discussed in this work.

main comments

1. It would help if you explained the remax objective more clearly in the introduction with the help of a small example. The two armed bandit example could have been mentioned here: and it could be stated that if we took M=2 samples and obtained the outcome w.p. 0.25 twice, then the remax objective for selecting arm 1 would be 0, and selecting arm 0 would be 1. While this may seem very trivial, providing a small example would really help in understanding for someone who has not seen the remax objective before.
It would also help to specify that the inner expectation in eq (1) is necessary to capture the case when we use a stochastic policy.

2. The fundamental idea of the remax objective seems similar to risk aware approaches, where you have some temperature $\lambda$ that controls whether we are risk seeking or risk aware (similar to this idea in the sense that you still have to select the hyper $M$ which acts like a temperature for exploration). Are the authors aware of some significant differences between the two lines of research?

3. Should the summation term be written outside the max in eq (6) ? Since you are summing over all $m$, taking max over it has no meaning. The summation index could be changed, or the summation could be taken to the left for clarity. Or just use brackets to be precise.

4. In the equation in line 250 col 2: why is it important to use the $max(x, 0)$ expression? The highest possible $q_{A_m}$ value should give a gap that is >= 0 because $W_{-m}$ would give the second largest q-value. So why is it necessary to use the $max(x, 0)$ expression? While it doesn't seem to affect the correctness of the equation, I'm not sure how it helps here.

5. In the experiments, whenever you use the ReMax PPO update, you should be getting good performance for the ReMax objective. This shouldn't necessarily imply an improvement in the original RL objective. Even when you show improvement by using ReMax for the bandit case in Figure 1 and 2, you are showing the features of ReMax on the ReMax objective and not the RL objective. From that point of view, I don't entirely understand why you should expect improvement in results in terms of the original RL objectives as shown in the experiments. I understand the point of view of exploration to help in certain environments, but then towards the end of optimization when exploration is "complete", should there be an exploitation step where we anneal the value of $m$ towards $1$ to ultimately optimize for the original RL objective?

6. Is this approach capable of working for the sparse environments in the Atari benchmark, for example, Montezuma's Revenge?

7. The results seem convincing particularly for the MinAtar benchmark, but this is just a smaller benchmark. Over Craftax and the selective Atari experiments, the performance seems to be somewhat comparable across some of the existing baselines. This makes it hard to concretely reach the judgement that using this type of objective is actually helping in those latter two benchmarks. I do think that the entropy graphs are convincing, but I also want to re-emphasize the point 5 that I made. Perhaps at the end of this training, if we anneal the value of $m$ towards the original RL objective, then we might have better performance. I would be interested to hear the authors' thoughts on this.

minor comments

1. line 128 column 1: shouldn't this be written as a set {0, 1} instead of the interval (0, 1)?

2. In line 267 col 2: is the blue term the expected improvement for the M-th draw or for all the M draws collectively? If it is just the M-th draw, how is the effect of the M-1 draws factored in to begin with?

typos

1. equation at line 250 col 2: I think the first part should be W_{-j} instead of W_{-m} ?

---

> ### Author Rebuttal · Authors · 2026-03-31
>
> Thank you very much for the review.
>
> > For comments 1
>
> We agree that a small example would improve accessibility. We will add a two-armed bandit example in the introduction to illustrate that with $M=2$, pulling the same arm twice does not improve the best observed reward, whereas sampling both arms can. We will also clarify that the inner expectation in Eq. (1) is needed to handle stochastic policies.
>
>
> > For comment 2
>
> We agree that ReMax and risk-sensitive objectives are superficially similar, but conceptually different. In many risk-sensitive RL formulations, the objective changes how the return distribution is evaluated, for example through entropic criteria such as $\frac{1}{\lambda}\log \mathbb{E}[\exp(\lambda G)]$ [1, 2] to encode preferences over variability or tail behavior. By contrast, ReMax evaluates the expected maximum over multiple sampled actions, thereby encouraging exploration.  We will clarify this distinction in the related work section.
>
>
> > For comment 3
>
> Thank you for pointing this out.
> You are right that the summation should be written outside the maximization for clarity.
> We will revise the expression accordingly.
>
>
>
> > For comment 4
>
> The $(x)_+$ form is not introduced to change the objective, but to enable the per-sample policy-gradient form in Proposition 4.1. The naive gradient in Eq. (6) entangles the $M$ sampled actions through the max, making a single-trajectory unbiased estimator difficult. As shown in App. B.2 (Eq. 25–26), the ReLU form, together with the symmetry of the i.i.d. samples, separates the contribution of each action. We will clarify this point in the camera-ready version.
>
>
> > For comment 5
>
> We agree that the main benefit of ReMax is its exploration-promoting effect. At the same time, as shown in the deterministic bandit case (Fig. 1, right), **once uncertainty is removed, ReMax still converges to the optimal deterministic policy. Thus, annealing $m$ toward $1$ is not theoretically necessary to recover the original RL objective**.
> Practically, however, annealing may still be useful: in MinAtar, larger $m$ maintains higher entropy, but overly large $m$ can reduce final performance. We therefore view annealing as an interesting direction for future work.
>
>
>
> > For comment 6
>
> Sparse-reward exploration, such as Montezuma’s Revenge, is outside our current scope, so we do not expect the current RePPO implementation to perform well there. Our experiments focus on stochastic exploration rather than deep exploration in extremely sparse-reward settings. We agree this is an important future direction, and combining ReMax with curiosity bonuses or explicit uncertainty modeling may make it more suitable for such environments.
>
>
>
> > For comment 7
>
> Thank you for the positive assessments on the MinAtar results. On this point, we would like to emphasize that the MinAtar results are actually very strong as mentioned in the response to Reviewer 3Ltc.
>
> For Craftax and the selected Atari experiments, we agree that the gains are more mixed.
> Our interpretation, however, is that these results still support a central claim of the paper: RePPO consistently increases policy entropy during training, **while the return benefit depends on how much the environment benefits from sustained stochastic exploration**.
> In some Atari games, less exploratory variants perform better, such as PPO without entropy regularization or RePPO with smaller retry parameters (e.g., $m=0.9$ or $1.0$).
> In contrast, in Craftax, exploration appears more beneficial: RePPO with $m=1.2$ performs much better than PPO without entropy and is competitive with PPO+entropy and PPO+RND.
> We therefore view these results not as contradicting the method, but as showing that RePPO exposes a controllable exploration-exploitation tradeoff whose return impact is environment-dependent.
>
> Annealing $m$ could potentially further improve this tradeoff in some cases.
> However, since less exploratory variants already perform better in part of the Atari suite, we believe this would require a more careful study than we could include here.
>
>
>
> > minor comment 1
>
> We will fix the part accordingly.
>
> > minor comment 2
>
> The blue term is the expected improvement of a single draw, with the $M$-th draw fixed to action $a$. The effect of the other $M-1$ draws is already incorporated through $W_{M-1}$, the maximum over those draws.
>
> > typo
>
> The notation is correct: $m$ is fixed first, and $W_{-m}$ is the maximum over all samples except $A_m$. Hence, for any $j\neq m$, $q_{A_j}\le W_{-m}$, so
> $\max_{j\in[M]}(q_{A_j}-W_{-m})=(q_{A_m}-W_{-m})_+$.
>
>
>
> **Reference:**
> [1] Howard, Ronald A., and James E. Matheson. "Risk-sensitive Markov decision processes." Management science 18.7 (1972): 356-369.
> [2] Fei, Yingjie, et al. "Exponential bellman equation and improved regret bounds for risk-sensitive reinforcement learning." Advances in neural information processing systems 34 (2021): 20436-20446.

---

> > ### Author Rebuttal · Reviewer_wtaF · 2026-04-03
> >
> > Thank you for providing such a detailed response to each of my points.
> >
> > Regarding the distinction between risk-aware approaches and ReMax, I would also add that based on point 5: the risk aware objectives would need annealing to obtain the same objective as the risk-neutral one, whereas the ReMax objective does not require this (based on your explanation).
> >
> > Regarding point 6, it is completely understandable. That question was more out of curiosity and environments like Montezuma's requires additional techniques for long term exploration that is out of the reach of 'basic' RL approaches.
> >
> > Regarding point 7, yes the MinAtar results are convincing. While the other 2 benchmarks are not as convincing, they are still more on the positive side. I did not think that the other 2 benchmarks are contradicting the hypothesis set by the paper, rather I believed they were just mild indicators. I'm also inclined to agree with the point 1 in weaknesses made by reviewer Zv2H, although I think this is challenging to show and so I maintain that the current results are satisfactory. Due to these inherent limitations in the experiments, but overall satisfactory outcome, I would like to maintain my current mild positive score.

---

> > > ### Author Response · Authors · 2026-04-07
> > >
> > > Thank you for your follow-up and for maintaining your positive assessment. We are glad that our rebuttal addressed your concerns. We also appreciate your helpful perspective on the distinction from risk-aware objectives and on the scope of the empirical evaluation. We will reflect these points in the final version.

---

### Official Review · Reviewer_Zv2H · 2026-03-12

**Soundness:** 2
**Presentation:** 3
**Significance:** 3
**Originality:** 3
**Overall Recommendation:** 4
**Confidence:** 3

**Summary:**

This paper proposes ReMax, a reinforcement learning objective that evaluates a policy by the expected maximum return over M hypothetical samples drawn from the policy, while accounting for epistemic uncertainty over Q-values. The key insight is that exploration should emerge naturally from the retry mechanism. The authors derive a closed-form computation of the ReMax objective using a telescoping decomposition over sorted Q-values, develop an Expected Improvement (EI) based policy gradient that is estimable from single-trajectory returns, and generalize the integer retry count M to a continuous parameter m for fine-grained exploration control. Experiments on MinAtar and Craftax show that RePPO maintains higher policy entropy and achieves competitive or superior performance compared to PPO with entropy bonuses and RND, without using any explicit exploration bonus.

**Compliance With Llm Reviewing Policy:**

Affirmed.

**Final Justification:**

The authors have effectively addressed my concerns. The additional clarification of the experimental setup strengthens the paper. I am pleased to support acceptance.

**Key Questions For Authors:**

Please see the weaknesses, additionally,
1. On the Atari hard-exploration games (Appendix F), performance peaks at m=0.9–1.0, meaning less exploration is beneficial, the empirical gains are sometimes contradictory to the exploration narrative?

**Limitations:**

yes

**Strengths And Weaknesses:**

## Strengths
1. The core formalization is clean. It adapts exploration intensity to the magnitude of reward uncertainty, as demonstrated clearly in the Bernoulli bandit example where Softmax remains flat across variance levels while ReMax increasingly favors the high-variance arm.
2. The mathematical development is thorough and practical. The telescoping decomposition for computing expected max (Proposition 3.2) and EI (Proposition 4.3) converts expensive sampling-based computation into O(K log K) closed-form expressions. The generalization from integer M to continuous m is natural and enables fine-grained exploration control that would be impossible with integer retries alone.
3. The algorithm design is simple and well-ablated. The new algorithm modifies PPO only through the advantage computation, requiring no auxiliary models, no additional reward signals, and minimal computational overhead.

## Weaknesses
1. My biggest concern is whether RePPO functions primarily as a convergence rate controller rather than a principled uncertainty-adaptive mechanism. The paper frames ReMax as an uncertainty-adaptive exploration mechanism that should explore more when uncertain and less when confident. This property is demonstrated cleanly in the bandit setting with a proper posterior. However, in deep RL, the paper relies on implicit uncertainty from critic nonstationarity without providing evidence that this adaptation actually occurs. The entropy curves show RePPO maintains higher entropy overall, but do not show that entropy is high specifically in states where the agent is uncertain and low where it is confident. Without such analysis, it is difficult to distinguish RePPO's behavior from simpler alternatives that also maintain higher entropy, such as entropy regularization with a decaying coefficient (which would also converge to the true optimum), or simply using a smaller learning rate to slow convergence.

2. The experimental baselines are narrow. The paper positions ReMax as an alternative to bonus-based exploration and extensively discusses other methods (curiosity-based methods, count-based methods, etc)  in the related work. Yet the only dedicated exploration baseline is RND.

3. The scope is limited to small discrete action spaces. The closed-form computation requires enumerating and sorting Q-values for all actions, which is infeasible for continuous or large discrete spaces.

---

> ### Author Rebuttal · Authors · 2026-03-31
>
> Thank you very much for the review.
>
> > My biggest concern is whether RePPO functions primarily as a convergence rate controller rather than a principled uncertainty-adaptive mechanism ...
>
> We would first like to clarify that, as shown in the bandit experiments, **ReMax can promote exploration through two mechanisms**: it adapts to epistemic uncertainty when uncertainty is modeled explicitly (Figure 1 Left and Center), and, even with deterministic rewards, larger \(M\) delays convergence and thus sustains exploration longer (Figure 1 Right). As discussed in the paper, our deep-RL instantiation may benefit from both effects.
>
> In deep RL, we use critic non-stationarity as an implicit uncertainty signal. While prior work (e.g., “The Phenomenon of Policy Churn”, https://arxiv.org/pdf/2206.00730) suggests this may be meaningful, we agree that it is still unclear how faithfully this captures epistemic uncertainty, or which mechanism is more dominant. We leave this to future work; more explicit uncertainty estimation may help ReMax adapt more directly to epistemic uncertainty.
>
> At the same time, even if the dominant effect in the current implementation is closer to convergence control, **we believe this still supports one of the intended mechanisms of ReMax rather than undermining it.** Empirically, we consistently observe that \(m>1\) maintains higher entropy in MinAtar and Atari, supporting our main claim that ReMax promotes exploration in deep RL.
>
> Regarding the alternatives you mentioned, a decaying entropy bonus is not a simple substitute, since it introduces another schedule that must be designed and tuned. Using a smaller learning rate is also not equivalent: the learning rate is already tuned, and simply reducing it does not generally improve performance.
>
>
> > The experimental baselines are narrow. The paper positions ReMax as an alternative to bonus-based exploration and extensively discusses other methods (curiosity-based methods, count-based methods, etc) in the related work. Yet the only dedicated exploration baseline is RND.
>
> We chose RND because it is a widely used exploration baseline and is also used in the official baseline implementations for Craftax. More importantly, the main purpose of our deep-RL experiments was to isolate whether **ReMax itself can sustain exploration without an auxiliary bonus**, using policy entropy as the main diagnostic. For this reason, our primary comparisons were against entropy regularization.
>
>
> > The scope is limited to small discrete action spaces. The closed-form computation requires enumerating and sorting Q-values for all actions, which is infeasible for continuous or large discrete spaces.
>
> We agree that the current implementation is restricted to discrete action spaces in which Q-values can be evaluated for all actions, and we already acknowledge this limitation in the paper. This assumption enables the closed-form computation, and the sorting step itself is not the main bottleneck since it only costs $O(K\log⁡K)$. The main challenge is instead full-action Q evaluation in very large discrete or continuous action spaces. As noted in the paper, ReMax can in principle be extended beyond this exact closed-form setting using sampling-based estimation, and we mention recent work in language modeling as one example of this direction.
>
> > On the Atari hard-exploration games (Appendix F), performance peaks at m=0.9–1.0, meaning less exploration is beneficial, the empirical gains are sometimes contradictory to the exploration narrative?
>
> We agree with this observation. In Appendix F, we evaluated the dense-reward ``hard-exploration'' Atari suite of Bellemare, et al. [1]. Empirically, these environments did not benefit from stronger stochastic exploration: performance peaked around $m=0.9$, $1.0$, and PPO without entropy regularization often outperformed more exploratory variants. We do not view this as contradictory to our exploration narrative. Rather, we interpret it as evidence that these tasks require relatively little additional exploration. This interpretation is also consistent with prior work suggesting that, in many Atari hard-exploration games proposed by [1], bonus-based exploration methods such as RND do not substantially improve over  $\epsilon$-greedy [2]. In this sense, our results further support the view that stronger exploration is not universally beneficial in these environments. **At the same time, RePPO still behaves consistently with our proposed mechanism**: larger $m$ maintains higher policy entropy, whereas smaller $m$ performs better when less exploration is needed.
>
>
> **Reference:** [1] Bellemare, Marc, et al. "Unifying count-based exploration and intrinsic motivation." Advances in neural information processing systems 29 (2016).
> [2] Taiga, Adrien Ali, et al. ``On bonus-based exploration methods in the arcade learning environment.'' arXiv preprint arXiv:2109.11052 (2021).

---

> > ### Author Rebuttal · Reviewer_Zv2H · 2026-04-06
> >
> > Thank you for providing the detailed responses, my concerns have been addressed and I'll increase my score.

---

> > > ### Author Response · Authors · 2026-04-07
> > >
> > > Thank you for your constructive comments and positive feedback. We are glad that our rebuttal could address your concern. In the final version, we will revise the paper so that the scope and intention of the empirical validation are clearer.

---

### Official Review · Reviewer_AWpV · 2026-03-12

**Soundness:** 3
**Presentation:** 4
**Significance:** 3
**Originality:** 3
**Overall Recommendation:** 5
**Confidence:** 3

**Summary:**

This paper introduces a new reinforcement learning objective,  Remax, which evaluates a policy through the expected maximum return over
$M$ samples instead of the standard expected return, thereby incorporating return uncertainty into the objective itself. One of the main results of the paper is that this objective naturally encourages exploration, without the need for explicit exploration bonuses. The authors further derive the gradient of this objective, which makes it possible to design variants of policy gradient and PPO specifically tailored to the Remax framework. These methods are then tested on several reinforcement learning environments, and the experiments support the claim that they indeed exhibit the expected exploratory behavior.

**Compliance With Llm Reviewing Policy:**

Affirmed.

**Final Justification:**

The authors successfully addressed all my concerns. I support the acceptance of this paper, which I believe will be of great interest to the reinforcement learning and policy gradient communities.

**Key Questions For Authors:**

See Weakness section.

**Limitations:**

yes

**Strengths And Weaknesses:**

## Strenghs

1) The paper proposes a new objective that directly accounts for exploration in reinforcement learning. A key strength of the approach is its simplicity in terms of hyperparameterization, since it only introduces a single additional parameter. Overall, I find the idea both elegant and clearly worthwhile.

2) The paper is very well written and easy to follow.

3) The derivations are rigorous, and proofs are provided for all stated results.

4) The experiments confirm the usefulness of this framework, as RePPO is competitive with the considered baselines and even outperforms them while requiring fewer hyperparameters to tune.

## Weaknesses

1) When looking at the hyperparameters used for training PPO, I noticed that the entropy coefficient is only explored for two very small values. Since increasing this coefficient would likely encourage more exploration and increase the entropy of the learned policy, this choice may underestimate the performance of the PPO baseline. I would therefore encourage the authors to rerun the experiments with a wider sweep over the entropy coefficient, for example
$$\{ 10^{-3}, 10^{-2}, 10^{-1}, 1\}.$$

For the rebuttal, it would not be necessary to repeat all experiments across every environment; rerunning the comparison on one or two representative environments would already help clarify whether the proposed method remains competitive against a better-tuned PPO baseline

2) The algorithm requires sorting the Q-values, which introduces an additional computational overhead.

3) The related work section is missing the following papers, which discuss the use of alternative divergences beyond entropy to encourage improved exploration:

[1] Labbi et al, Beyond Softmax and Entropy: Convergence Rates of Policy Gradients with f\-SoftArgmax Parameterization $\&$ Coupled Regularization,
The Fourteenth International Conference on Learning Representations, 2026.

[2] Zhao et al, Towards a Sharp Analysis of Learning Offline \$f\$-Divergence-Regularized Contextual Bandits,
The Fourteenth International Conference on Learning Representations, 2026

4) (Minor typo): I think that Table 2 should not include the entropy coefficient row, since RePPO does not rely on an entropy regularizer.

---

> ### Author Rebuttal · Authors · 2026-03-31
>
> Thank you very much for the review.
>
> > When looking at the hyperparameters used for training PPO, I noticed that the entropy coefficient is only explored for two very small values…
>
> To address this concern, we additionally swept the entropy coefficient of PPO-Q over $\{0, 10^{-3}, 10^{-2}, 10^{-1}, 1\}$ and evaluated the resulting policy entropy and final return on MinAtar, Atari, and Craftax.
> Overall, while larger entropy coefficients ($10^{-1}$ and $1$) indeed increased policy entropy, they did not improve performance over our original hyperparameter settings with entropy coefficients $0$ and $10^{-2}$.
> In addition, none of the newly added settings ($10^{-3}, 10^{-1}, 1$) outperformed the original best.
> **Thus, this additional sweep does not change our main conclusion that RePPO remains better than or competitive against a better-tuned PPO baseline.**
> Please also note that $10^{-2}$ is the default entropy coefficient in already tuned public implementations (PGX [1] for MinAtar, CleanRL [2] for Atari, and the official Craftax baseline implementation [3] ).
> We provide the results for MinAtar in the tables below, and, for the rest of the environments (Atari and Craftax), we provide the results in the anonymous GitHub repository (http://anonymous.4open.science/r/icml2026-rebuttal-materials-3F27/README.md).
>
> **Entropy**
>
> | Env | PPO-Q (ent=0) | PPO-Q (ent=0.001) | PPO-Q (ent=0.01) | PPO-Q (ent=0.1) | PPO-Q (ent=1.0) | RePPO(1.2) | RePPO(1.4) |
> | --- | --- | --- | --- | --- | --- | --- | --- |
> | Breakout | 0.21 | 0.20 | 0.33 | 0.76 | 1.03 | 0.59 | 0.73 |
> | Asterix | 0.71 | 0.75 | 0.92 | 1.41 | 1.59 | 1.23 | 1.37 |
> | Space Invaders | 0.32 | 0.34 | 0.53 | 1.12 | 1.36 | 0.77 | 0.96 |
> | Freeway | 0.10 | 0.11 | 0.16 | 0.63 | 1.08 | 0.38 | 0.62 |
>
> **Return**
>
> | Env | PPO-Q (ent=0) | PPO-Q (ent=0.001) | PPO-Q (ent=0.01) | PPO-Q (ent=0.1) | PPO-Q (ent=1.0) | RePPO(1.2) | RePPO(1.4) |
> | --- | --- | --- | --- | --- | --- | --- | --- |
> | Breakout | 105.00 | 82.29 | 98.09 | 102.61 | 9.74 | **167.83** | 155.06 |
> | Asterix | 31.83 | 32.09 | 35.52 | 40.69 | 16.48 | 42.14 | **42.51** |
> | Space Invaders | 557.47 | 476.48 | 572.23 | 491.12 | 45.91 | 672.39 | **706.81** |
> | Freeway | 64.74 | 64.77 | 65.50 | 65.12 | 55.26 | **66.20** | 65.28 |
>
>
> > The algorithm requires sorting the Q-values, which introduces an additional computational overhead.
>
> The additional operation introduced by RePPO is sorting the Q-values, which costs $O(K \log K)$ in the number of actions and is independent of the retry parameter.
> In practice, this overhead is small relative to the rest of training.
> As shown in Fig.10 (MinAtar) and Fig.16 (Craftax), RePPO is very close in training time to PPO-Q, which is the most relevant comparator since it also uses a Q-critic.
> It is particularly clear in Craftax, where PPO-V+RND incurs noticeably higher computation due to the additional RND networks, while RePPO remains comparable to PPO-Q.
>
>
> > The related work section is missing the following papers, which discuss the use of alternative divergences beyond entropy to encourage improved exploration:
>
> We agree that Labbi et al.(2026) is a relevant missing reference, and we will add it to the related work.
> Their paper studies policy optimization beyond the standard softmax/entropy pairing by using alternative $f$-divergence-induced parameterizations and coupled regularization, which is clearly relevant to our discussion of alternatives to entropy-based exploration.
> We also appreciate the pointer to Zhao et al.(2026).
> While that paper is less directly about exploration methods per se, since it focuses on the theory of offline $f$-divergence-regularized contextual bandits, it is still relevant as broader context on $f$-divergence regularization.
> We will add the discussion in the related work section.
>
>
> > For typo
>
> Thank you for pointing out, we fill fix in the next revision.
>
> **Reference:**
> [1] https://github.com/sotetsuk/pgx
> [2] https://github.com/vwxyzjn/cleanrl
> [3] https://github.com/MichaelTMatthews/Craftax_Baselines/tree/main

---

> > ### Author Rebuttal · Reviewer_AWpV · 2026-04-02
> >
> > The authors have fully addressed my concerns. The additional experiments further strengthen the paper and make the contribution more convincing. I therefore support acceptance and would be very pleased to see this paper accepted at ICML.

---

> > > ### Author Response · Authors · 2026-04-07
> > >
> > > Thank you for your constructive comments and positive feedback. We are glad that our rebuttal helped address your concern. We will revise the manuscript based on your comments. In particular, we will expand the related work section to include work on exploration based on f-divergence.

---

### Official Review · Reviewer_3Ltc · 2026-03-13

**Soundness:** 3
**Presentation:** 3
**Significance:** 3
**Originality:** 3
**Overall Recommendation:** 4
**Confidence:** 4

**Summary:**

This paper proposes ReMax, an objective that scores a policy by the expected maximum return over M samples under epistemic uncertainty, thereby inducing stochastic exploration without explicit bonuses. The authors derive a policy gradient estimable from single-trajectory returns, generalize the retry count to a continuous parameter m, and implement RePPO, a PPO variant. Experiments on MinAtar and Craftax show that RePPO promotes exploration without bonus terms.

**Compliance With Llm Reviewing Policy:**

Affirmed.

**Final Justification:**

The authors adequately addressed all of my concerns and I maintain that this is a nice contribution with higher confidence.

**Key Questions For Authors:**

See weaknesses

**Strengths And Weaknesses:**

Strengths
- The conceptual contribution is interesting. The idea that exploration arises from greedy optimization of a retry-based objective under uncertainty is, to my knowledge, novel.
- The policy gradient derivation is the main technical result of the paper. The progression from the naive estimator through the EI-based single-action form (Proposition 4.1) to the closed-form computation (Proposition 4.3) is interesting.
- The paper is well-written. For example, the bandit analysis in Section 2 is an effective warmup.
- The continuous generalization from integer M to real m > 0 is practically useful, enabling fine-grained exploration-exploitation control.

Weaknesses
- My main concern is with the Q-uncertainty modeling, which to me could be improved and further discussed. The practical algorithm treats the nonstationarity of the critic during training as a proxy for epistemic uncertainty over Q-values, but doesn't this conflate other sources of uncertainty with epistemic uncertainty?
- The experimental evaluation feels too narrow for the breadth of the claims. For a paper whose central claim is about exploration, the absence of hard-exploration benchmarks is bit of a gap.
- I'm not sure the comparison baselines are strong enough. The paper compares against PPO variants and PPO-V+RND, but not against more sophisticated exploration methods like bootstrapped DQN, or other posterior-sampling approaches, maybe more recent epistemic NN papers.

---

> ### Author Rebuttal · Authors · 2026-03-31
>
> Thank you very much for the review.
>
> > My main concern is with the Q-uncertainty modeling, which to me could be improved and further discussed…
>
> Thank you for pointing this out. We agree that our current uncertainty modeling may not be complete. However, even in the current method, **we do not believe this conflates epistemic uncertainty with aleatoric uncertainty**, because the Q-value already integrates over aleatoric uncertainty by taking an expectation over returns. This is also part of the reason why, in RePPO, we use Q-values rather than directly emulate the retry mechanism (e.g., via simulator resets). We will clarify this point in the revision.
>
> Regarding more sophisticated uncertainty modeling, we have already noted in the limitation part that our current treatment can be improved, for example through more explicit uncertainty modeling. We nevertheless appreciate this comment and will expand the limitations paragraph. For example, we can add the following clarification: “In the bandit setting, we confirmed that ReMax can adapt to uncertainty over returns. However, in our current deep-RL instantiation, we do not explicitly model Q-uncertainty. To fully realize the benefits suggested by the bandit results, it would be promising to combine ReMax with methods that explicitly quantify uncertainty, such as Bayesian methods ”
>
> We also care about whether the current method truly captures epistemic uncertainty, and about what the best way to estimate it would be. We leave this to future work because this is the first paper on ReMax, and our main goal here is to introduce the idea, analyze its properties in a controlled setting, and provide a practical deep-RL instantiation that demonstrates its potential.
>
> > The experimental evaluation feels too narrow for the breadth of the claims. For a paper whose central claim is about exploration, the absence of hard-exploration benchmarks is bit of a gap.
>
> In the deep-RL experiments, our goal is not to claim that ReMax solves all sparse-reward or deep-exploration problems. Rather, **our empirical claim is that ReMax/RePPO promotes stochastic exploration**, as evidenced by consistently higher policy entropy during training without any explicit exploration bonus.
>
> **This is a central claim of the experimental section, and also stated in the introduction.** From that perspective, the fact that $M>1$ reliably increases entropy is itself an important result, while the return improvements, especially the clear gains on MinAtar, show that this additional exploration can be practically useful.
>
> > I'm not sure the comparison baselines are strong enough. The paper compares against PPO variants and PPO-V+RND, but not against more sophisticated exploration methods like bootstrapped DQN, or other posterior-sampling approaches, maybe more recent epistemic NN papers.
>
> Our main goal in the deep-RL experiments was to isolate whether ReMax itself can promote exploration, as reflected in higher policy entropy, without introducing a separate exploration bonus. For this reason, we primarily compared against PPO-based baselines and standard bonus-based methods.
>
> That being said, we do not believe the current baselines are weak.
> **On MinAtar, our results compare favorably to prior published results.** For example, PQN appears to report previous state-of-the-art results on this benchmark (https://arxiv.org/pdf/2407.04811v2, Fig. 8), and our tuned PQN already exceeds those values, while RePPO further improves over tuned PQN in mean return. On Breakout in particular, RePPO reaches about 175, whereas the PQN paper reports around 75. Another recent reference, the RLC Outstanding Paper Award winner (https://openreview.net/pdf?id=LZAafvwVMa), reports scores of roughly 25, 12, 45, and 110 on Asterix, Breakout, Freeway, and Space Invaders at 5M steps, while our corresponding results at 5M steps are about 30, 50, 60, and 200, and improve further at 10M steps. The Revisiting Rainbow paper (https://arxiv.org/pdf/2011.14826) also reports weaker MinAtar results than ours.
>
> For completeness, we also considered adding recent approximate-sampling / Langevin-based exploration methods [1,2]. In the larger-scale Atari games, we share three environments with them (Alien, H.E.R.O., and Qbert). On these overlapping games, however, our PPO baseline is already competitive with or stronger than the reported performance of those methods while using fewer frames. We therefore did not expect adding these baselines to materially change the main empirical takeaway of the current draft. That said, we agree that a broader comparison to posterior-sampling and uncertainty-estimation methods would be valuable in a more comprehensive follow-up study.
>
> **Reference:**
> [1] Ishfaq et al., Provable and Practical: Efficient Exploration in Reinforcement Learning via Langevin Monte Carlo, ICLR 2024
> [2] Ishfaq et al., More Efficient Randomized Exploration for Reinforcement Learning via Approximate Sampling, RLC 2024

---

> > ### Author Rebuttal · Reviewer_3Ltc · 2026-04-03
> >
> > Thanks for the responses. I continue to think that this is a nice contribution and maintain my score of 4.

---

> > > ### Author Response · Authors · 2026-04-07
> > >
> > > Thank you for your constructive comments and positive feedback. We are glad that our rebuttal could address your concern. In the final version, we will revise the manuscript based on your comments. In particular, we will clarify our discussion of uncertainty quantification.

---

### Decision · Program_Chairs · 2026-04-30

**Decision:**

Accept (regular)

**Comment:**

The paper presents a technically solid and conceptually elegant contribution to reinforcement learning by showing how exploration can emerge from a retry-based objective. Reviewers were unanimous in their appreciation of the mathematical progression, from the bandit warmup to the closed-form policy gradient, and the practical utility of the continuous retry parameter $m$. While initial concerns were raised regarding the implicit nature of the uncertainty modeling in deep RL and the breadth of the experimental evaluation, the authors' rebuttal successfully addressed these points through additional hyperparameter sweeps and clarifying the scope of their stochastic exploration claims. The consensus among reviewers is that the work is original, well-written, and likely to influence future research in policy gradient methods. Consequently, the paper is recommended for acceptance.